# Localized epigenetic silencing of a damage-activated WNT enhancer limits regeneration in mature *Drosophila* imaginal discs

Robin E Harris, Linda Setiawan, Josh Saul[†], Iswar K Hariharan*

Department of Molecular and Cell Biology, University of California, Berkeley, Berkeley, United States

**Abstract** Many organisms lose the capacity to regenerate damaged tissues as they mature. Damaged *Drosophila* imaginal discs regenerate efficiently early in the third larval instar (L3) but progressively lose this ability. This correlates with reduced damage-responsive expression of multiple genes, including the WNT genes *wingless (wg)* and *Wnt6*. We demonstrate that damage-responsive expression of both genes requires a bipartite enhancer whose activity declines during L3. Within this enhancer, a damage-responsive module stays active throughout L3, while an adjacent silencing element nucleates increasing levels of epigenetic silencing restricted to this enhancer. Cas9-mediated deletion of the silencing element alleviates WNT repression, but is, in itself, insufficient to promote regeneration. However, directing *Myc* expression to the blastema overcomes repression of multiple genes, including *wg*, and restores cellular responses necessary for regeneration. Localized epigenetic silencing of damage-responsive enhancers can therefore restrict regenerative capacity in maturing organisms without compromising gene functions regulated by developmental signals.

*For correspondence: ikh@ berkeley.edu

Present address: [†]Massachusetts Institute of Technology, Cambridge, United States

Competing interests: The authors declare that no competing interests exist.

## Introduction

The ability of tissues to regenerate following damage varies greatly between different species (*Li et al., 2015*). Even some vertebrate species are capable of fully regenerating organs that have essentially no regenerative capacity in humans such as heart tissue in adult zebrafish (*Poss et al., 2002*) and limbs in urodele amphibians such as newts and salamanders (*Tanaka and Reddien, 2011*). Understanding the cellular differences between homologous organs that differ in their regenerative capacity among species could suggest genetic or pharmacological manipulations that improve the regenerative capacity of damaged organs in humans.

The ability to regenerate specific organs differs not only between species, but can also change within a single species during its development. In general, for organisms that have determinate growth, i.e. growth that ceases once a genetically pre-determined size is reached (*Sebens, 1987*), the capacity for regeneration usually decreases with increasing maturity. For example, embryonic and neonatal mice are able to regenerate myocardial tissue following substantial damage (*Drenckhahn et al., 2008*; *Porrello et al., 2011*). However, within the first week of life, this capacity to regenerate is lost almost completely, and the damaged tissue is instead replaced by fibrotic scarring (*Porrello et al., 2011*). Other examples include appendage regeneration in developing *Xenopus* (*Dent, 1962*; *Slack et al., 2004*) and the progressive loss of digit regeneration in mice and humans (*King, 1979*; *Borgens, 1982*; *Reginelli et al., 1995*).

**eLife digest** The ability of many animals to regenerate damaged tissues decreases as they age, for example, newborn mice can regenerate damaged heart tissue while older mice cannot. Researchers are trying to discover why older animals lose the ability to regenerate, which may help us to develop therapies that can regenerate damaged tissues in humans.

Fruit flies are relatively simple animals that are often used as models in biology experiments. In young fruit fly larvae, there are tissues called imaginal discs that regenerate well after damage; however these discs lose this ability as the larva matures. A gene called *wingless* is very active in young larvae if the imaginal discs become damaged and helps them to regenerate. Previous studies show that this gene is not as strongly switched on in older larvae after tissue damage. However, since *wingless* also performs other roles in fruit flies, it is not clear how cells can stop *wingless* from being highly activated after tissue damage without affecting other important processes.

Regions of DNA called enhancers can regulate the activity of genes. Harris et al. studied an enhancer that had previously been shown to drive the activation of *wingless* following tissue damage. The experiments show that there are two separate sections within the enhancer that control *wingless* activity. One section activates *wingless* in response to tissue damage and can perform this role even as the tissue matures. However, in older larvae, the other section alters the properties of the first section's DNA to reduce its effectiveness. By switching off this section of the enhancer – but not the *wingless* gene itself – the activity of *wingless* no longer responds to tissue damage, but can still be regulated by signals that influence other processes.

Harris et al. also found that several other genes that are not active in mature tissues can be re-activated by a protein called Myc. Therefore, increasing the production of Myc in cells can promote the regeneration of more mature tissues. The next step is to find out if other genes involved in fly tissue regeneration are regulated in a similar way. A future challenge will be to find out if the same mechanism also limits tissue regeneration in humans and other more complex animals as they age.

The basis for this phenomenon is not well understood. In some instances where tissue re-growth is dependent on stem cell based replacement of tissue (for review see *Tanaka and Reddien, 2011*), the loss of a stem cell population could account for the inability to regenerate. However, this explanation cannot account for situations where regeneration is not driven by a stem cell population but rather by the proliferation and de-differentiation of adjacent tissue. In these cases the loss of regenerative capacity in a still-developing organism necessitates a mechanism that selectively inactivates regenerative processes while still allowing cell proliferation and differentiation associated with normal development. Since most genes that function during regeneration also have a variety of functions either in normal development or in maintaining homeostasis in the same tissue, it is presently unclear how their expression could be regulated so as to selectively block regeneration.

Genetic studies using *Drosophila* have provided important insights into the genetic regulation of tissue growth. Many of these studies have examined growth of the imaginal discs, larval epithelial tissues that are precursors of adult structures, such as the wing and the eye (*Cohen, 1993*). Imaginal discs are capable of regenerating missing portions following damage (*Worley et al., 2012*). Studies of imaginal disc regeneration were pioneered by the group of Ernst Hadorn and were mostly conducted by transplanting damaged discs into the abdomens of adult female flies (*Ursprung and Hadorn, 1962*). More recently, imaginal disc regeneration has been studied in intact larvae by damaging portions of the disc via brief expression of a pro-apoptotic gene in a spatially-restricted manner (*Smith-Bolton et al., 2009*; *Bergantiños et al., 2010*). Using either a genetic ablation system (*Smith-Bolton et al., 2009*), or following X-ray irradiation (*Halme et al., 2010*), it was observed that the capacity of the wing-imaginal disc to regenerate progressively diminished during the later stages of the third larval instar as the larva approached the beginning of metamorphosis. Expression of *wingless (wg, Drosophila WNT1* ortholog) is robustly upregulated in regenerating discs following genetic ablation. However, in more mature discs, which do not regenerate, there is a strong reduction in this *wg* upregulation (*Smith-Bolton et al., 2009*).

We chose to address the basis for reduced regeneration in mature discs by examining the mechanisms underlying the reduction in *wg* upregulation following damage. The upregulation of WNT proteins is an early and often essential response to tissue damage, observed in diverse species including *Hydra* and planaria (*Gurley et al., 2008*; *Petersen and Reddien, 2008*; *Lengfeld et al., 2009*), as well as vertebrate species such as zebrafish, axolotls and *Xenopus* (*Kawakami et al., 2006*; *Stoick-Cooper et al., 2007*; *Yokoyama et al., 2007*; *Lin and Slack, 2008*). Multiple studies have demonstrated that WNT signaling is essential for regeneration and is able to augment the process (*Kawakami et al., 2006*; *Stoick-Cooper et al., 2007*; *Yokoyama et al., 2007*; *Lin and Slack, 2008*), even in tissues that do not normally undergo regeneration (*Kawakami et al., 2006*). Thus, the mechanisms that link tissue damage in imaginal discs to *wg* upregulation, and the basis for diminished *wg* upregulation with increasing maturity, are likely to provide insights into similar processes in diverse organisms.

Here we characterize the properties of an enhancer in the major WNT locus of *Drosophila* (*Schubiger et al., 2010*) that mediates the upregulation of both *wg* and *Wnt6* following damage to imaginal discs, and show that deletion of this enhancer impairs regeneration. We show that damage-responsive activation and age-dependent attenuation can each be ascribed to separate modules within the enhancer, and while the damage-responsive module is equally effective in promoting damage-responsive gene expression in mature discs, its activity is overridden by an adjacent element that nucleates epigenetic silencing with increasing effectiveness as the larva matures. Importantly, this epigenetic silencing is restricted to the immediate vicinity of this enhancer, thereby not interfering with the activation of *wg* and *Wnt6* by distinct developmentally regulated enhancers. We also describe a way of overcoming this silencing and improving regeneration in more mature larvae.

## Results

In order to study the mechanisms that regulate regenerative growth in imaginal discs, we used a genetic ablation system developed in our laboratory (*Smith-Bolton et al., 2009*), which demonstrated that damage-responsive *wg* expression in mature discs is reduced compared to younger discs. A pro-apoptotic gene, either *eiger (egr)* or *reaper (rpr)*, is expressed specifically in the wing pouch by *rn-Gal4*, and temporally regulated by temperature-controlled inactivation of Gal80 (encoded by *Gal80$^{ts}$*) (*Figure 1A*). Depending on the transgene used, this system is hereafter referred to as *rn$^{ts}$>egr* or *rn$^{ts}$>rpr*. This system was used to initiate cell death early in the third larval instar (L3) on day 7 (hereafter "day 7 discs") or late in L3 on day 9 (hereafter "day 9 discs") (*Figure 1A*). After 20 hr of *rpr* expression or 40 hr of *egr* expression, the tissue that comprises the wing pouch is almost completely ablated in both a day 7 and a day 9 disc (*Smith-Bolton et al., 2009*). However, ablations initiated later in L3 result in significantly less regenerative growth, yielding adult flies with wings of greatly reduced size (*Figure 1B*). Regeneration of younger discs is associated with an extended larval stage (*Figure 1C* and *Figure 1—figure supplement 1*), which allows sufficient time for tissue re-growth before the pupal stage of development. This extension is significantly smaller in ablated day 9 discs (*Figure 1C* and *Figure 1—figure supplement 1*). Importantly, the decline in regenerative growth is not simply due to restricted proliferation in mature discs, as both ablated and unablated discs from day 9 discs still express markers of cell-cycle progression, albeit at a lower level than in day 7 discs (*Figure 1—figure supplement 2A–C*). As suggested by the reduced damage-responsive *wg* expression in older discs, other mechanisms likely exist to limit regenerative growth.

To examine the basis for a progressive loss of regenerative capacity, we compared the morphological characteristics of *rn$^{ts}$>rpr* ablated day 7 and day 9 discs using light sheet microscopy (*Figure 1D–E*, *Videos 1* and *2*). Since this technique involves embedding the specimens in agarose for visualization, the three dimensional architecture of the disc is preserved and can be accurately examined. Additionally, we used confocal microscopy to visualize the cells at the edge of the ablated region (*Figure 1F–I*). The JNK signaling pathway is known to be activated strongly in regenerating imaginal discs (*Bosch et al., 2005*; *Mattila et al., 2005*; *Bergantiños et al., 2010*). We therefore used an AP-1 reporter (*Chatterjee and Bohmann, 2012*) to identify the cells in which the JNK pathway was active. In both day 7 discs and day 9 discs we observed that the disc proper (DP) epithelium in the central portion of the wing pouch had been ablated, while both the peripodial epithelium (PE) and the basement membrane underlying the DP were intact (*Figure 1D–E*). Indeed,

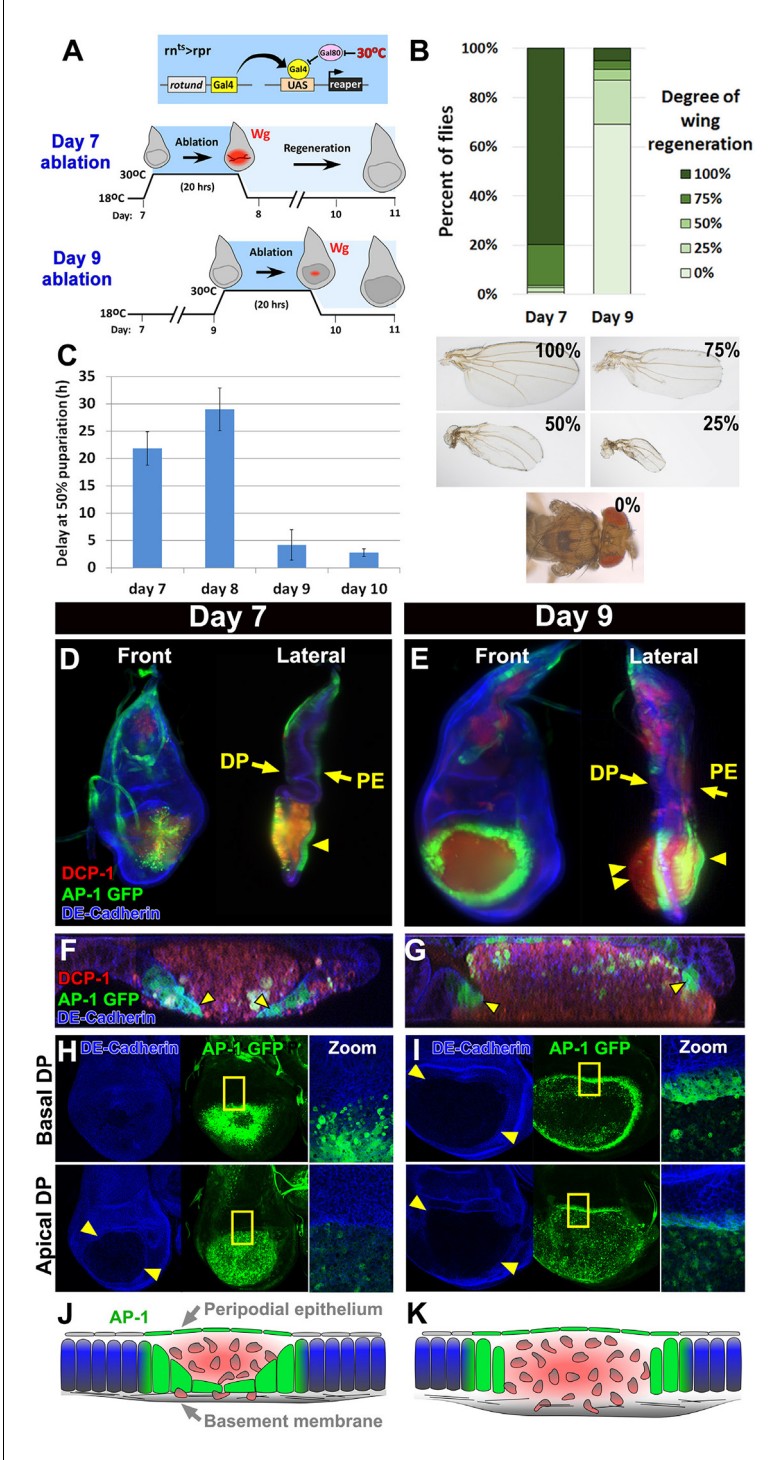

**Figure 1.** Cellular changes that accompany reduced regeneration in older discs. (**A**) The *rn^{ts}>rpr* ablation system used to induce tissue damage and stimulate regeneration in L3 wing-imaginal discs on day 7 and day 9 of development. The molting from 2nd to 3rd larval instar occurs around day 6 at 18°C. Gal4 under the control of the *rotund (rn)* enhancer drives the pro-apoptotic gene *rpr* (or *egr* in *rn^{ts}>egr*) in the developing wing pouch. Ablation is repressed at 18°C by a ubiquitous temperature-sensitive Gal80 (Gal80^{ts}), until the temperature is increased to 30°C to inactivate the Gal80^{ts} and induce cell death. Ablation of the pouch is complete after 20 hr for *rn^{ts}>rpr* (or 40 hr for *rn^{ts}>egr*). The temperature is then reduced (downshift) to end ablation and allow regeneration. Wg (red) is observed in response to ablation in a day 7 disc, but this response is greatly diminished in a day 9 ablated disc, correlating with a loss of regenerative ability. (**B**) Distribution of adult wing sizes from *rn^{ts}>rpr* flies where a 20 hr

*Figure 1 continued on next page*

*Figure 1 continued*

ablation was initiated on day 7 or day 9 of larval development, showing a decrease of regeneration in mature discs. n > 400 flies per experiment. Examples of wing sizes scored are shown below. In almost all cases both wings of a single fly showed equivalent levels of regeneration. (C) Average delay measured as hours at which 50% of larvae have pupariated compared to unablated wild type. Ablation induces a developmental delay of 22.2 hr on day 7, which is progressively reduced to just 2.8 hr on day 10. Error bars are SD of at least 3 biological repeats. (D-I) Wing discs imaged following *rpr* ablation at the time of the downshift to 18°C. (D-E) Light sheet microscopy of wing discs following ablation with *rn^{ts}>rpr* on day 7 (D) and day 9 (E). Discs were embedded in agarose and imaged in 360°. Two orthogonal planes are shown. Abundant cellular debris is observed between the peripodial epithelium (PE) and the bulging basement membrane of the disc proper (DP) and is visualized with an antibody to cleaved *Drosophila* DCP1 (DCP1, red). In day 9 discs cell debris is seen outside of the DP, confined by the basement membrane (double arrowhead). Cells expressing the AP-1 reporter (AP-1, green) are found adjacent to the ablated region in the DP and in the intact PE overlying the ablated region (single arrowheads) on both day 7 and day 9 ablations. Blue: DE-cadherin. (F-G) Confocal Z-sections of the ablated pouch of a day 7 (F) and day 9 (G) ablation. There is a discontinuity in the DP following ablation. However following a day 7 ablation (F), but not following a day 9 ablation (G), AP-1 positive cells (arrowheads) appear to migrate inward to close the gap in the DP. (H-I) Expression of an AP-1 GFP reporter in discs ablated on day 7 (H) and day 9 (I), and imaged at basal (top panels) and apical (bottom panels) levels of the DP. Zoom panel shows enlargement of the area within the yellow rectangle. On both days 7 and 9, AP-1-GFP is also observed in the cells of the DP surrounding the ablated region and in the debris within the DP. The flattened cells that seem to be closing the ablated region on day 7 are observed at basal but not apical focal planes (H). The AP-1 positive cells adjacent to the ablation in day 9 discs retain their columnar morphology (G) and do not seem to cover the discontinuity in the DP caused by the ablation (I). (J-K) Drawings illustrating the contrasting cellular responses to ablations initiated on day 7 and day 9.

The following figure supplements are available for figure 1:

**Figure supplement 1.** Developmental delay associated with damage declines in mature discs.

**Figure supplement 2.** Cells in the wing disc are still proliferating at a time when they can no longer regenerate.

---

visualizing the basement membrane using a GFP-tagged version of Viking (Vkg) (*Buszczak et al., 2007*), a collagen that is deposited in the basement membrane, shows no lack of continuity (*Figure 1—figure supplement 2D*). The lumen between the basement membrane and the overlying PE of each disc was filled with cellular debris, marked by the activated caspase DCP-1 (*Figure 1D–E*). In both day 7 and day 9 discs the AP-1 reporter was strongly expressed in the cells adjacent to the ablated portion of the DP (*Figure 1H–I*), as well as the overlying PE (*Figure 1D–E*, single arrowheads), indicating that the JNK pathway is activated consistently in discs of both ages.

However, despite the equivalent level of activation, the pattern of AP-1 expression differed between the two developmental stages. In day 7 discs the cells expressing the AP-1 reporter in the DP became flattened and appeared to migrate towards the interior of the ablated region (*Figure 1F*), leaving the majority of the debris between the PE and the newly migrated epithelium. Hence, these AP-1 positive cells could be seen clearly in basal confocal sections but not in apical ones (*Figure 1H*). In contrast, in day 9 discs, the AP-1 positive cells at the edge of the ablated region maintained their columnar profile (*Figure 1G*) and remained at the periphery of the ablated wing pouch (*Figure 1I*), allowing the debris to extend beyond the plane of the basal DP (*Figure 1E*, double arrowheads). Thus, the cellular response to damage is markedly different in mature discs compared to younger discs (*Figure 1J–K*).

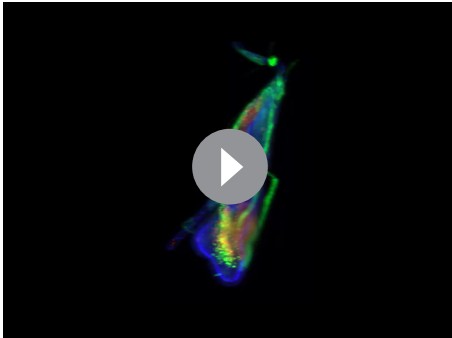

**Video 1.** The morphological response of a day 7 disc to ablation. Three dimensional imaging using light sheet microscopy of discs ablated with *rn^{ts}>rpr* on day 7, with AP-1-GFP (green) and stained for DCP-1 (red), and DE-Cadherin (blue). A 360° rotation of the disc is shown, and the AP-1-GFP channel is shown separately.

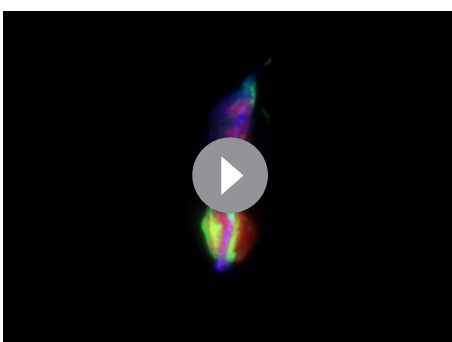

**Video 2.** The morphological response of a day 9 disc to ablation. Three dimensional imaging of a *rn^{ts}>rpr* ablated day 9 disc, imaged as in *Video 1*.

This altered response of mature discs could result from differences in the expression of genes that are induced by ablation. As shown previously (*Smith-Bolton et al., 2009*), both *wg* and *Myc* are upregulated following ablation on day 7, but this upregulation is reduced considerably on day 9 (*Figure 2A–B*, *Figure 2—figure supplement 1*). We also compared the activity of other signaling pathways that are known to function in the context of imaginal disc damage and regeneration (*Figure 2C–G*). For all genes, we examined whole-disc expression by imaging at the level of the basal and apical DP, and the PE of each disc (*Figure 2—figure supplement 1*, *Figure 2—figure supplement 2*, *Figure 1—figure supplement 2B*). A reporter of *Stat92E* activity (*Bach et al., 2007*), an indicator of signaling via the JAK/STAT pathway, which has recently been implicated in imaginal disc regeneration (*Katsuyama et al., 2015*), was strongly upregulated all around the ablated region on day 7, but on day 9 the expression was restricted to those regions that also express the reporter in unablated discs of a comparable developmental stage (*Figure 2C*, *Figure 2—figure supplement 1*). Yki activity has been shown to be increased in regenerating discs (*Grusche et al., 2011*; *Sun and Irvine, 2011*). The *bantam*-GFP reporter (*Matakatsu and Blair, 2012*), which can be activated by increased Yki activity, also shows a moderate decline in expression following ablation as larvae progress through L3 (*Figure 2D*, *Figure 2—figure supplement 1*). The cell cycle regulator Cyclin E and a reporter of E2F activity also showed a similar reduction in ablation-induced expression in a day 9 disc (*Figure 1—figure supplement 2B–C*), indicating that the proliferative response to damage can still occur in late stage discs, but is diminished. Thus a number of damage-responsive signaling pathways show reduced activation following tissue damage in day 9 discs when compared to day 7 discs (summary, *Figure 2G*).

As the JNK pathway is essential for regeneration (*Bosch et al., 2005*; *Mattila et al., 2005*; *Bergantiños et al., 2010*), we also examined the expression of several JNK target genes in day 7 and day 9 discs. Our observations of the AP-1-GFP reporter suggest that the JNK pathway is still strongly active in mature ablated discs (*Figure 1E,G,I*, *Figure 2—figure supplement 2*). Surprisingly however, when we examined expression of *matrix metalloprotease 1 (MMP1)*, a JNK target gene that is important for cell migration (*Page-McCaw et al., 2003*) and wound healing (*Stevens and Page-McCaw, 2012*), we found a strongly reduced response to ablation in a day 9 disc compared to that of a day 7 (*Figure 2E*). This was indicated by both protein levels (*Figure 2E*) and transcriptional readout by an *MMP1-lacZ* reporter (*Uhlirova and Bohmann, 2006*) (*Figure 2—figure supplement 2*). The JNK phosphatase *puckered (puc)*, another JNK target gene (*Ring and Martinez Arias, 1993*), was also not as strongly activated in an ablated day 9 disc (*Figure 2—figure supplement 2*). Additionally, expression of *DILP8*, a secreted protein released by damaged tissue that delays entry into metamorphosis (*Colombani et al., 2012*; *Garelli et al., 2012*), and which is regulated by JNK signaling (*Colombani et al., 2012*; *Katsuyama et al., 2015*), was reduced in day 9 discs when compared to day 7 discs (*Figure 2F*, *Figure 2—figure supplement 2*). Thus, while JNK activation, as assessed by a synthetic promoter with multimerized AP-1 binding sites, is comparable on day 7 and day 9, a number of known JNK pathway targets demonstrate strongly reduced damage-responsive expression in mature discs compared to younger discs. This correlates with an altered morphological response to damage and a loss of regenerative capacity. One explanation for this change in gene expression is that the damage-responsive signaling cascades that activate these pathways are less active in day 9 discs, although the AP-1 reporter suggests that, at least for JNK targets, this is not the case. An alternate possibility is that the transcriptional response to these activating signals is actively repressed by a mechanism that becomes more potent as discs mature. To distinguish between these possibilities, we conducted a detailed characterization of the mechanisms that regulate *wg* expression following damage, as *wg* is a gene whose orthologs have well documented roles in regeneration in many organisms.

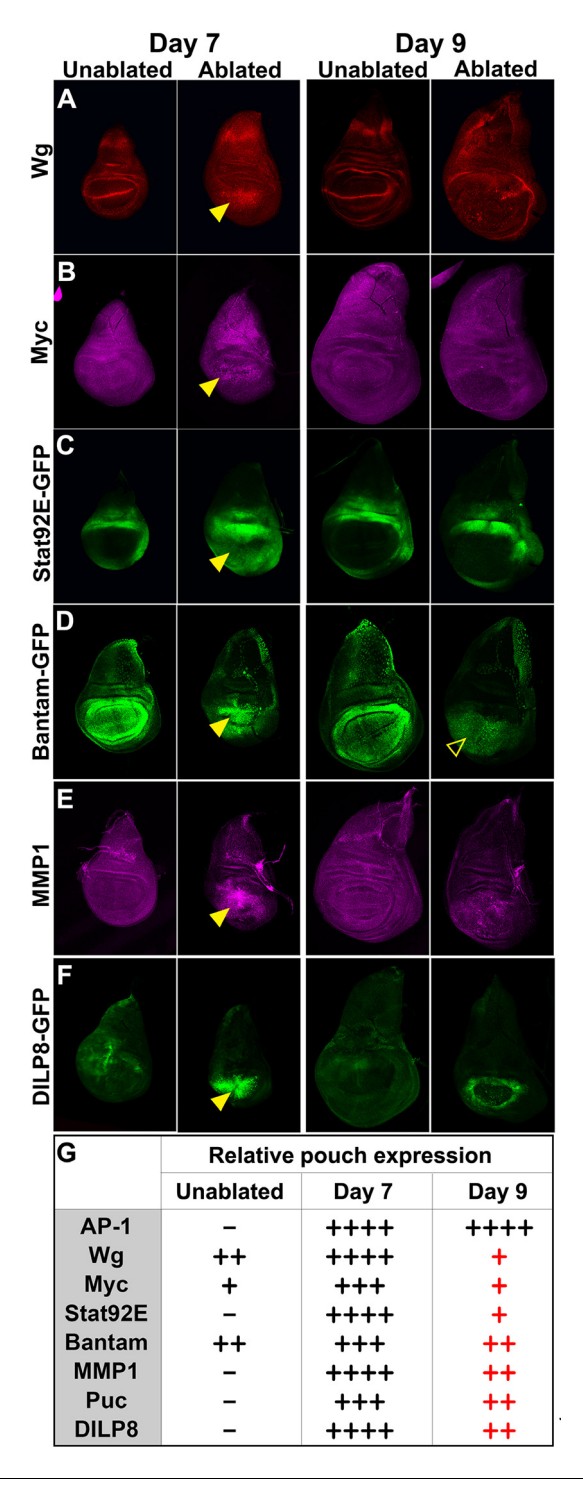

| G | Relative pouch expression | | |
|---|---|---|---|
| | Unablated | Day 7 | Day 9 |
| AP-1 | − | ++++ | ++++ |
| Wg | ++ | ++++ | + |
| Myc | + | +++ | + |
| Stat92E | − | ++++ | + |
| Bantam | ++ | +++ | ++ |
| MMP1 | − | ++++ | ++ |
| Puc | − | +++ | ++ |
| DILP8 | − | ++++ | ++ |

**Figure 2.** Changes in expression of key regulators of regeneration that accompany reduced regeneration in older discs. (A-F) Basal confocal sections of discs ablated with *rn*$^{ts}$*>rpr* on day 7 or day 9 imaged at the downshift to 18°C, with equivalently staged unablated discs, detecting (A) Wg, (B) Myc, (C) *Stat92E-GFP* reporter (D) *bantam-GFP* reporter, (E) Mmp1 and (F) *DILP8-GFP* reporter. In each case the expression of the protein or reporter is increased by ablation on day 7 (arrowheads) when compared to identically stained unablated control discs. In discs ablated on day 9, there is no detectable upregulation of Wg (A) or Myc (B) in the pouch. Expression of the *bantam-GFP* (D), Mmp-1 (E) and *DILP8-GFP* (F) is consistently reduced when compared to a day 7 ablation and coincides mainly with cellular debris (for example, open arrowhead in D). (G) Summary of gene expression
*Figure 2 continued on next page*

*Figure 2 continued*

observed in this study in discs ablated on day 7 and day 9, compared to unablated discs. Results that change between day 7 and day 9 are indicated in red.

The following figure supplements are available for figure 2:

**Figure supplement 1.** Expression of reporters shown at multiple sections in the epithelium.

**Figure supplement 2.** JNK pathway target gene expression shown at multiple levels.

## The BRV118 enhancer activates both *wg* and *Wnt6* expression to promote regeneration

A number of enhancers have been identified that regulate the normal pattern of *wg* expression in the wing disc during L3 (*Neumann and Cohen, 1996*; *Pereira et al., 2006*; *Koshikawa et al., 2015*). However, *wg* expression following damage appears to be regulated by a distinct enhancer. Examination of genomic DNA fragments spanning the entire *WNT* gene cluster, including the *wg* locus, by Schubiger and colleagues (*Schubiger et al., 2010*) identified a ~3 kb region, named BRV118, that was activated in imaginal discs following mechanical injury. In these studies, BRV118 was assumed to primarily regulate *wg*. However, since BRV118 is located in the middle of a cluster of WNT genes, between *wg* and *Wnt6* (*Figure 3A*), it is possible that it might regulate multiple WNT genes.

To investigate this possibility further, we used RNA in situ hybridization to examine wing imaginal discs following genetic ablation using *rn^{ts}>egr* for changes in expression of each of the four WNT genes located near BRV118. In the absence of tissue ablation, expression patterns of all four genes on day 7 matched previously published descriptions (*Figure 3B*), including low ubiquitous expression of *Wnt10* (*Janson et al., 2001*), and a *wg*-like pattern for *Wnt4* and *Wnt6* (*Gieseler et al., 2001*; *Doumpas et al., 2013*), with *Wnt4* being weaker and more diffuse, as previously reported (*Gieseler et al., 2001*). In day 7 *rn^{ts}>egr* discs, *wg* RNA was detected at high levels around the damaged tissue, as was *Wnt6* RNA (*Figure 3B*). Neither *Wnt4* nor *Wnt10* expression was appreciably elevated. Moreover, the upregulation following damage of both *wg* and *Wnt6* RNA was diminished in day 9 discs (*Figure 3C*). Thus, our findings indicate that expression of both *wg* and *Wnt6*, the two genes that flank BRV118, is upregulated in response to damage, and that this response declines with larval maturation.

To study whether BRV118 regulates the expression of both *wg* and *Wnt6*, we examined their expression in ablated *wg^1* discs. The BRV118 element is located approximately 8 kb downstream of the *wg* gene and largely overlaps the region that is deleted in the *wg^1* allele (*Figure 3D*). Homozygous *wg^1* adults are viable, and have the ability to develop normally sized and patterned wings, suggesting this region is not essential for normal development. However, a proportion of flies exhibit a variably-penetrant wing-to-notum transformation, reflecting a semi-redundant requirement of this region early in larval development for wing-pouch specification by *wg* (*Sharma and Chopra, 1976*). This loss of wing tissue is phenotypically distinct and easily distinguishable from ablated wings (*Figure 3—figure supplement 1*). In ablated *wg^1* discs, expression of both *wg* and *Wnt6* was markedly reduced (*Figure 3E*) suggesting that sequences disrupted in the *wg^1* mutant are necessary for the upregulation of both genes. We therefore examined regeneration in *wg^1* flies following *rn^{ts}>egr* ablation. The size of adult wings after regeneration was considerably reduced in *wg^1* flies (*Figure 3F*). Thus, sequence alterations in the *wg^1* mutant, most likely the deletion of the BRV118 enhancer, compromise regeneration.

## Separate enhancer modules mediate damage response and age-dependent silencing

To further characterize the properties of the BRV118 enhancer, we constructed a reporter transgene consisting of a 2.9 kb enhancer fragment and a basal promoter driving eGFP expression (*Figure 4A*). In the absence of tissue damage, GFP expression was not detected in imaginal discs at any point during L3, or in any earlier larval stages that were examined, including L2 or L1 (*Figure 4B*, and data not shown). However, upon ablation with *rn^{ts}>egr*, GFP is strongly expressed

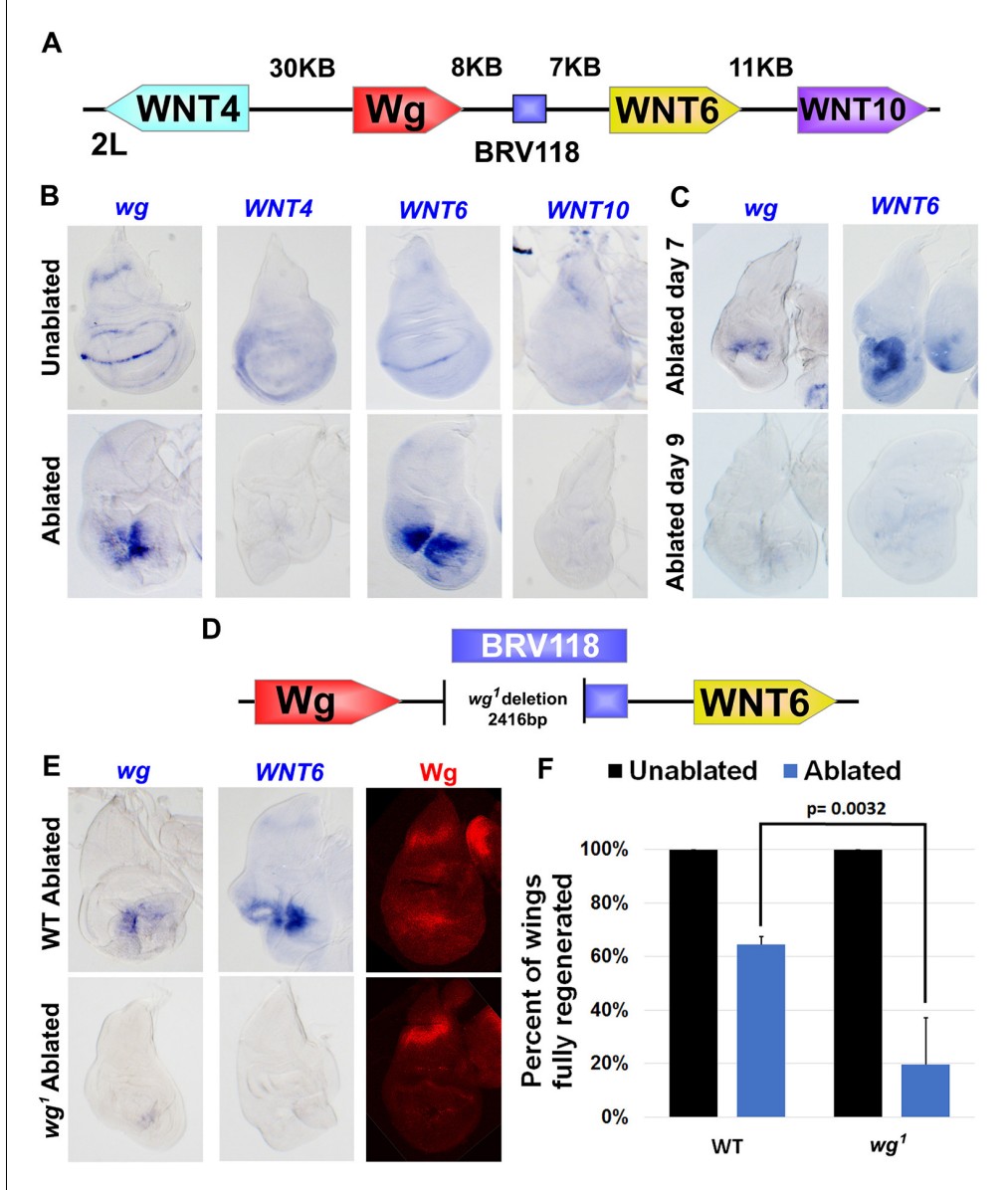

**Figure 3.** The BRV118 enhancer is necessary for *wg* and *Wnt6* expression following tissue damage. (**A**) Schematic of the major WNT locus showing the BRV118 enhancer (blue rectangle). Approximate intergenic distances are labeled. (**B**) RNA in situ hybridization to detect *wg, Wnt4, Wnt6* and *Wnt10* RNA in day 7 unablated (top row) and $rn^{ts}$>*egr* ablated (bottom row) wing discs. Only *wg* and *Wnt6* demonstrate significant upregulation of RNA in response to damage. (**C**) RNA in situ hybridization to detect *wg*, and *Wnt6* RNA in discs ablated with $rn^{ts}$>*egr* on day 7 and day 9. Transcription of both *wg* and *Wnt6* in response to damage is absent in a day 9 disc. (**D**) Schematic of the lesion in the $wg^1$ allele that deletes most of the BRV118 enhancer. (**E**) RNA in situ hybridization to detect *wg* and *Wnt6* RNA and Wg protein, in wild type (top row) and homozygous $wg^1$ (bottom row) day 7 $rn^{ts}$>*egr* ablated discs. The damage-specific expression of *wg* and *Wnt6* RNA, and Wg protein levels, are reduced in $wg^1$ discs compared to wild type. (**F**) Adult wing sizes following ablation with $rn^{ts}$>*egr* on day 7 in wild type and $wg^1$ homozygotes, indicating the percentage of animals that eclose with fully regenerated wings. Regeneration of $wg^1$ homozygous discs is significantly reduced compared to wild type. Error bars indicate SD of at least three independent experiments, scoring a total of >200 animals in each experiment. Only untransformed $wg^1$ wings were scored (for explanation of untransformed see *Figure 3—figure supplement 1*).

The following figure supplement is available for figure 3:

**Figure supplement 1.** The $wg^1$ allele causes transformation of wing to notum in a proportion of the population.

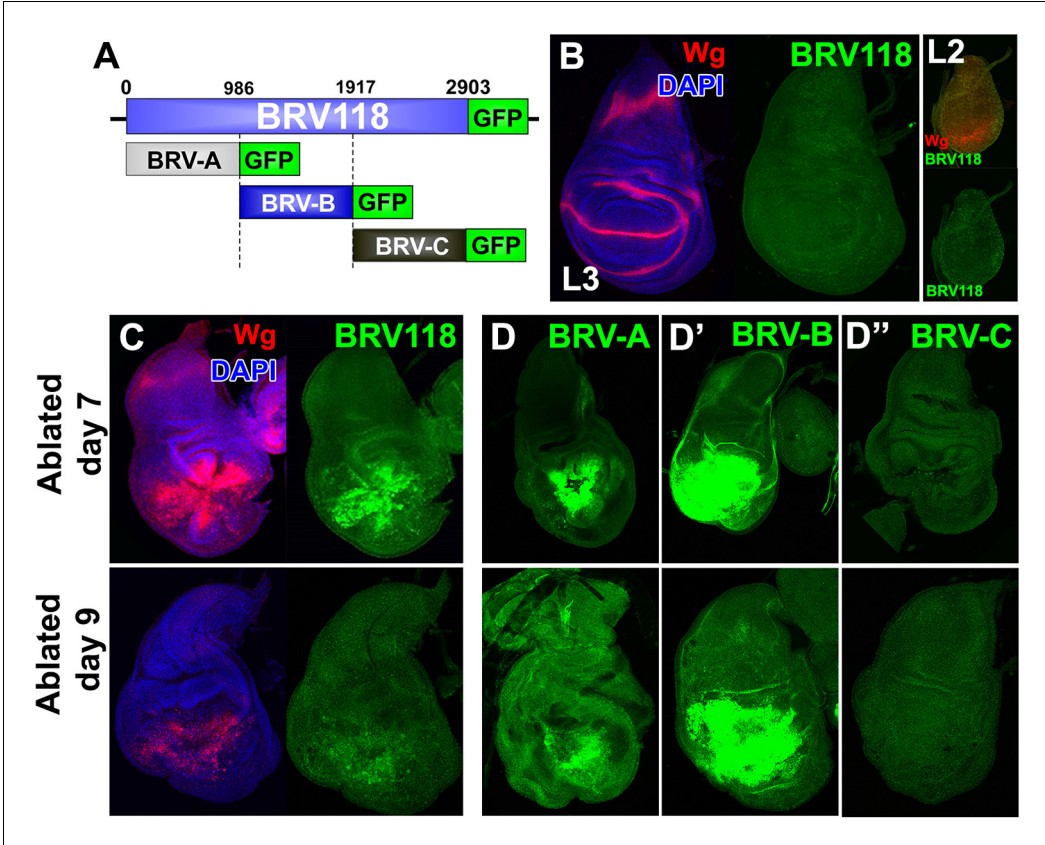

**Figure 4.** BRV118 is a damage-responsive WNT enhancer. (A) Schematic of the 2.9 kb BRV118-GFP reporter transgene, and the subdivisions used to generate the ~1 kb BRV-A, BRV-B and BRV-C reporters. All transgenes were inserted into the same genomic landing site to enable direct comparison. Numbers indicate nucleotide positions in BRV118. (B) Unablated day 7 disc bearing the BRV118-GFP reporter, stained for Wg (red) and DAPI (blue). No GFP is detected during normal development throughout L3 (green) or L2 (insets). Discs were staged using the Wg expression patterns: diffuse Wg in the primordial pouch tissue to indicate L2 and the defined D-V stripe and "hinge circles" to indicate L3 (red). (C, D-D'') Discs bearing the BRV118-GFP (C), BRV-A (D), BRV-B (D') or BRV-C (D'') reporters ablated with $rn^{ts}$>egr on day 7 (top row) or day 9 (bottom row). The BRV118-GFP reporter (green) is expressed in a pattern closely resembling that of damage-induced Wg (red), on both day 7 and day 9. The BRV-A reporter has weaker activation than the full enhancer on day 7 and on day 9. The BRV-B reporter has much stronger activation than the full BRV118 and is not attenuated with age. BRV-C is not expressed following ablation.

The following figure supplement is available for figure 4:

**Figure supplement 1.** The BRV118 enhancer reporter demonstrates damage-responsive but not developmentally-regulated *wg* expression.

in dying cells and in the surrounding tissues in a pattern that closely resembles damage-induced Wg expression (*Figure 4C*). Importantly, the reporter is also expressed at much higher levels in $rn^{ts}$>egr day 7 discs than in day 9 discs (*Figure 4C*). Thus, both the activation of damage-responsive expression of *wg*, as well as the reduced responsiveness with increasing maturity, are likely mediated in significant part at the transcriptional level. Moreover, the 2.9 kb BRV118 fragment must include sequences that can mediate both of these aspects of *wg* transcription.

The BRV118 reporter is also activated in $rn^{ts}$>rpr discs, though not as strongly (*Figure 4—figure supplement 1A–B*). Similar differences between $rn^{ts}$>egr and $rn^{ts}$>rpr discs were also observed with respect to Wg protein expression. During regeneration, Wg and GFP mostly overlap initially but their patterns of expression differ as regeneration proceeds (*Figure 4—figure supplement 1C*). Wg resolves into its characteristic developmentally-regulated pattern while GFP expression persists in a

broader separate domain within the regenerated tissue, distinct from dead cells (*Figure 4—figure supplement 1D*). Thus the 2.9 kb fragment contains sequences sufficient to drive reporter expression in a pattern characteristic of damaged and regenerating discs, yet lacks sequences necessary for directing the developmentally-regulated patterns of *wg* expression.

To identify damage responsive elements within the BRV118 enhancer, we generated further reporter constructs that each contained one of three non-overlapping fragments of the enhancer, each approximately 1 kb in length (*Figure 4A*). Of these, the BRV-B fragment directed strong damage-responsive reporter gene expression in both day 7 and day 9 discs (*Figure 4D'*). The BRV-A fragment also elicited some damage-responsive expression (*Figure 4D*), albeit far less than BRV-B, and the BRV-C fragment did not drive any detectable expression (*Figure 4D''*). The lack of age-dependent attenuation of expression driven by the BRV-B fragment indicates that damage-responsive transcription and its age-dependent attenuation can be dissociated, thereby implying that the signaling pathways that activate the damage-responsive element are fully active in imaginal discs of late L3 larvae.

## A damage-responsive module within the BRV118 enhancer is activated by many types of tissue injury

To test whether the BRV-B fragment functions as a damage-responsive enhancer in other contexts, we induced tissue damage in a variety of ways. Reporter activation was observed in and surrounding clones of cells that express the pro-apoptotic gene *egr* in wing, haltere, leg and eye discs (*Figure 5A–A'*, *Figure 5—figure supplement 1A–B*) indicating that the reporter can be activated by *egr*-induced cell death in most, if not all, imaginal discs. Similarly, inducing cell death by irradiation (*Figure 5B*) or physical fragmentation of discs in culture (*Figure 5C–D*) induced reporter gene expression. In all of these experiments, reporter activation was less evident following damage to the notum of the wing disc (*Figure 5A*, open arrowheads) or the larval brain (*Figure 5—figure supplement 1C*, open arrowheads) suggesting that additional mechanisms might regulate the tissue specificity of enhancer activation. Also, consistent with the differences between the full-length enhancer and the damage-responsive BRV-B module, irradiation or a physical cut does not induce Wg protein expression in mature discs, even though the BRV-B reporter is expressed (*Figure 5B,D*).

Wg has also been shown to be necessary for regeneration of the intestine in adult *Drosophila* following damage induced by the ingestion of cytotoxic agents (*Cordero et al., 2012*). In adults fed 5% dextran sulfate sodium (DSS), a sulfated polysaccharide that injures the intestinal epithelium (*Amcheslavsky et al., 2009*), the BRV-B reporter was activated in cells that have the morphology of enteroblasts and do not express *Delta*, which is a marker for intestinal stem cells (*Micchelli and Perrimon, 2006*; *Ohlstein and Spradling, 2007*) (*Figure 5E'*). Control animals fed the non-sulfated polymer dextran showed only scattered expression of GFP at low levels (*Figure 5E*). Thus the BRV-B enhancer responds to tissue damage induced in a variety of ways, including in situations that are known to activate *wg* expression in the context of regeneration.

## The damage-responsive enhancer is primarily activated by JNK signaling

In discs damaged in vivo by *rpr* genetic ablation or in culture by fragmentation, the spatial distribution of BRV-B activation correlates with the expression of the AP-1 reporter (*Figure 6A,F*). Additionally, as with the full length enhancer, BRV-B is activated more strongly by $rn^{ts}$>*egr* than by $rn^{ts}$>*rpr* (*Figure 6B–B'*). *egr* is thought to promote cell death in large part through activation of JNK signaling (*Igaki et al., 2002*), whereas blocking JNK signaling does not prevent *rpr*-mediated cell death (*Sun and Irvine, 2011*). These observations suggest that tissue injury could activate BRV-B in significant part via the JNK/AP-1 pathway. To test whether activation of JNK was sufficient to activate BRV-B driven expression, even in the absence of *egr*- or *rpr*-directed ablation, we expressed a constitutively active *hemipterous* transgene ($UAS$-$hep^{CA}$), which encodes an active form of JNK kinase. This strongly induced the reporter expression (*Figure 6C*). Conversely, in *hep* hemizygotes, where activity of JNK is greatly reduced, the upregulation of BRV-B following ablation with $rn^{ts}$>*rpr* is completely abolished (*Figure 6D–D'* ). To test whether the requirement for JNK signaling was disc autonomous and to rule out systemic influences, discs were dissected, physically fragmented and cultured in the presence of a small molecule JNK inhibitor. These discs failed to upregulate the BRV-

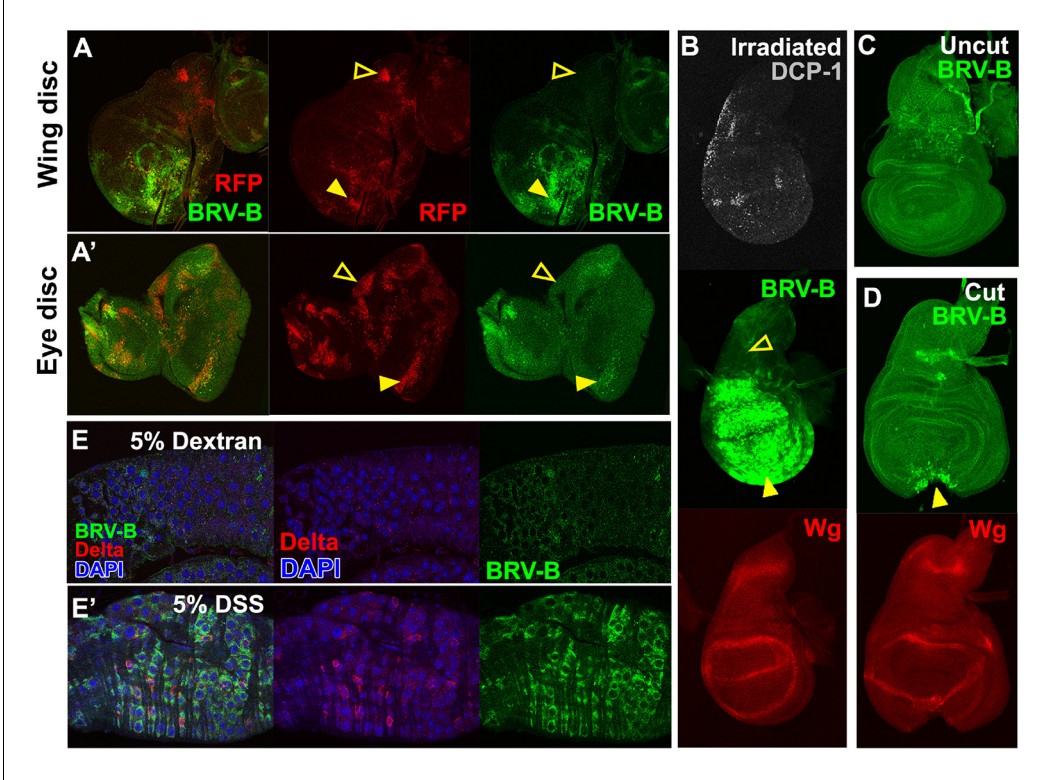

**Figure 5.** BRV-B is a damage-responsive enhancer in multiple contexts. (**A-A'**) *egr*-expressing clones activate BRV-B in some but not all regions of wing and eye discs. Heat-shock induced FLP-out clones expressing Gal4 (marked by RFP, red) were generated on day 4 and 5 of development in larvae bearing the BRV-B reporter. Clones were allowed to grow, and subsequently Gal4 was activated by inactivation of Gal80$^{ts}$ to express *egr* for 24 hr on day 8. GFP is expressed in and surrounding large dying clones (closed arrowheads) in wing discs (**A**) and eye/antennal discs (**A'**), but is absent from *egr*-expressing clones in various parts of each disc (open arrowheads), including the notum of the wing disc. (**B**) Disc from larva bearing the BRV-B reporter irradiated with 45 Gy on day 7, dissected after 16 hr and stained for cell death marker DCP-1 (gray), Wg (red) and GFP (green). The BRV-B reporter is strongly activated throughout the wing pouch and hinge (arrowhead), but not in the presumptive notum (open arrowhead). (**C-D**) Day 7 discs bearing the BRV-B reporter, physically cut (**D**, arrowhead indicates cut site) or left intact (**C**), and cultured in Schneider's medium for 16 hr, followed by staining to detect GFP and Wg. Reporter activation is detected specifically along the cut site. (**E-E'**) Confocal sections through midguts of adult flies bearing the BRV-B reporter following 2 days of feeding 5% dextran solution as control animals, (**E**) or gut-damaging 5% DSS solution (**E'**). Guts were stained with anti-Delta (red) to show the intestinal stem cell (ISC) population, and GFP to detect reporter activity. Damaged gut tissue in DSS fed animals increases ISC number. Strong induction of the BRV-B reporter (green) is observed but not in the population that expresses Delta.

The following figure supplement is available for figure 5:

**Figure supplement 1.** The BRV-B enhancer reporter is activated by genetic ablation in different tissues.

B reporter (*Figure 6—figure supplement 1*), indicating that JNK signaling in the wounded tissue itself is responsible for activation of the enhancer. Thus, at least under the conditions of these experiments, JNK signaling is necessary and potentially sufficient for BRV-B mediated expression.

The 931bp BRV-B fragment contains three strongly conserved high-consensus sites that match either the vertebrate or *Drosophila* AP-1 (Kay/Jra) motif (*Lee et al., 1987a*; *1987b*; *Perkins et al., 1988*) (*Figure 6E*). We deleted these sites in the BRV-B reporter to generate BRV-BΔAP1. Following physical fragmentation, discs bearing the BRV-B reporter strongly expressed GFP along the wound edges, coincident with AP-1-RFP (*Figure 6F*). In contrast, deletion of the AP-1 sites in BRV-BΔAP1 completely abolished expression of GFP in fragmented discs despite AP-1 activation (*Figure 6F'*). Similarly, the BRV-BΔAP1 reporter failed to be activated following *rn$^{ts}$>egr* ablation (*Figure 6G*), or

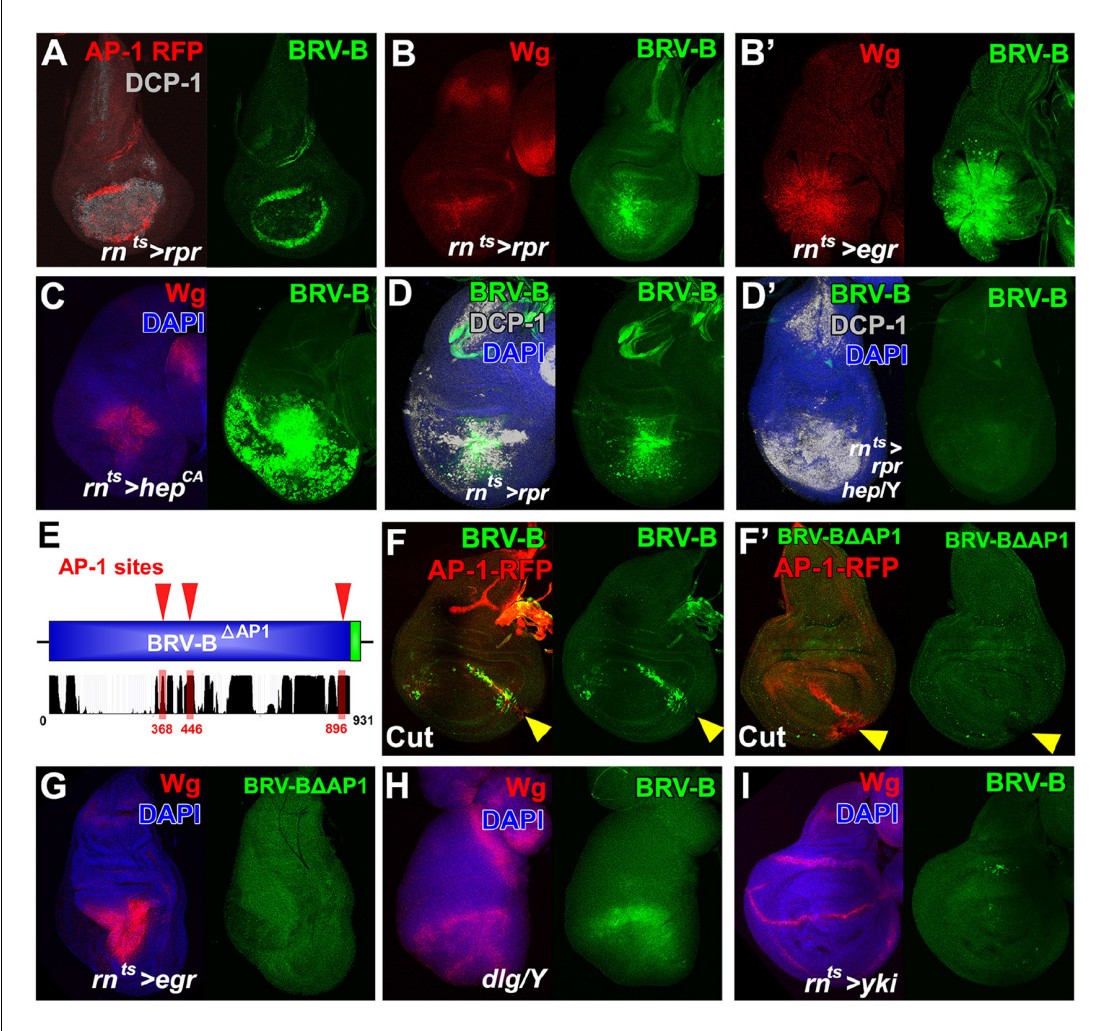

**Figure 6.** BRV-B reporter activation requires the JNK/AP-1 pathway. (**A**) Basal section of a day 9 disc bearing the AP-1 RFP reporter shows JNK pathway activity (red) and the BRV-B GFP reporter (green) following ablation by $rn^{ts}$>$rpr$, imaged at the time of downshift to 18°C. Reporter expression overlaps in wound edge cells, distinct from dead cells (DCP-1, gray), (**B-B'**) Day 7 discs bearing the BRV-B reporter, ablated using $rn^{ts}$>$rpr$ (B) or $rn^{ts}$>$egr$ (B') and imaged at the time of downshift to 18°C, demonstrating stronger expression of both Wg (red) and BRV-B GFP (green) following ablation with $egr$ compared to $rpr$. (**C**) A day 7 disc bearing the BRV-B reporter expressing $rn^{ts}$>$hep^{CA}$ for 20 hr to activate JNK signaling in the wing pouch. Both Wg (red) and the BRV-B GFP (green) are strongly expressed. (**D-D'**) Day 7 discs bearing the BRV-B reporter ablated with $rn^{ts}$>$rpr$ in a wild type (D) or $hep^-$ hemizygote (D'), imaged at the time of downshift to 18°C. GFP expression is abolished in the $hep^-$ mutant following ablation, while cell death is unaffected (DCP-1 staining, gray). (**E**) Schematic showing three predicted AP-1 binding sites (red) in the BRV-B enhancer fragment. The black histogram below shows a measure of the evolutionary conservation of the BRV-B DNA sequence in twelve *Drosophila* species, mosquito, honeybee and red flour beetle, based on a phylogenetic hidden Markov model (https://genome.ucsc.edu/). Numbers indicate nucleotide position in the BRV-B element. (**F-F'**) Day 7 discs bearing an AP-1-RFP reporter and BRV-B (F) or BRV-B with AP-1 sites deleted (BRV-BΔAP-1, F') physically cut and cultured for 16 hr. Activation of BRV-B (green) occurs along the edges of the cut site (arrowheads), coincident with AP-1-RFP expression (red) (F). Conversely the BRV-BΔAP-1 reporter is not activated by physical wounding (F'). (**G**) Basal section of a day 7 disc bearing BRV-BΔAP-1, following $rn^{ts}$>$egr$ ablation, imaged at the time of downshift to 18°C. Loss of the predicted AP-1 sites abolishes activation of the reporter (green). (**H**) Maximum projection Z-stack of a $dlg$ hemizygous mutant disc from early L3 bearing the BRV-B enhancer. BRV-B reporter expression is maximal in the dorsal hinge region coinciding with the site most prone to neoplastic overgrowth (***Khan et al., 2013***). (**I**) Maximum projection Z-stack of a day 7 disc bearing the BRV-B enhancer following expression of $rn^{ts}$>$yki$ for 20 hr. Hyperplastic growth resulting from $yki$ expression fails to activate BRV-B GFP expression.

The following figure supplements are available for figure 6:

**Figure supplement 1.** The BRV-B reporter is activated by physical damage, and is dependent on JNK signaling.

**Figure supplement 2.** Activation of the BRV-B reporter coincides with damage-induced, but not developmentally regulated JNK pathway activity.

after irradiation (data not shown). Together, these data indicate that upon tissue damage, WNT signaling is activated, in significant part, by JNK signaling via the BRV-B enhancer. Consensus sites for several other candidate activators are also present in the BRV-B sequence, including Grainyhead, a transcription factor that has been shown to function with AP-1 in larval epidermal wound repair (*Mace et al., 2005*), and the Dpp signaling transducers Mad and Medea. Several potential binding sites for the WNT pathway transcription factor TCF/Pangolin also exist, consistent with a previous observation that ubiquitously expressed *wg* can activate this enhancer (*Schubiger et al., 2010*). In this way, *wg* could activate or maintain its own expression as part of a feed forward mechanism during regeneration. Thus, although a high level of JNK signaling may be able to activate the enhancer on its own, under conditions of damage in vivo JNK signaling may act in concert with other damage-induced signals to activate *wg* expression via the BRV-B enhancer.

We also examined several other situations where JNK signaling is activated in the absence of tissue injury. The BRV-B reporter was expressed in discs mutant for *discs large (dlg)* (*Figure 6H*) or following knockdown of *scribble* (data not shown). In both situations there is a loss of epithelial apicobasal polarity that results in neoplastic growth. In contrast, expression of *yorkie (yki)*, the growth-promoting co-activator downstream of the Hippo pathway, for 20 hr using the $rn^{ts}$ system elicits overgrowth while preserving apico-basal polarity, but does not activate reporter expression (*Figure 6I*). The BRV-B reporter was also not activated in two situations where JNK is activated under physiological conditions: in the notum of late stage larval discs, which is required for dorsal thorax closure during the pupal stage (*Figure 6—figure supplement 2A*) and dorsal closure of the embryo (*Figure 6—figure supplement 2B*). These differences may reflect the extent of JNK activation since BRV-B is mostly activated under conditions where JNK is activated at high levels. Alternatively other factors specific to stress or damage may function in combination with JNK to modulate expression.

## A separate silencing element mediates maturity-dependent Polycomb-mediated repression

Unlike the 2.9 kb BRV118 full enhancer, the expression directed by BRV-B covers a broader region and is also unchanged between day 7 and day 9 (*Figures 4D'*, *7B*). Thus a negatively acting element must be present in either the BRV-A or BRV-C fragments, to reduce its area of expression and to mediate a further decrease in expression on day 9. Inclusion of the BRV-A fragment to the reporter (BRV-AB) elicited a pattern of expression that was similar to BRV-B (*Figure 7—figure supplement 1A*). In stark contrast, the addition of the BRV-C region to BRV-B (BRV-BC, *Figure 7A*) resulted in decreased expression in day 7 discs, both in the level and in the region of reporter expression (*Figure 7B*). Moreover, expression was greatly reduced in day 9 discs in a manner similar to the full-length 2.9 kb enhancer fragment (*Figure 7B*). Thus, a negatively acting element in BRV-C is responsible for limiting the response of BRV118 during regeneration and its ability to do so increases as the disc matures.

To map this element more precisely, we generated deletions from the end of BRV-C furthest away from BRV-B and examined their ability to silence GFP expression on day 9 (*Figure 7C*). We found that as more of the BRV-C fragment was deleted, there was a progressive loss of silencing, indicating that multiple regions within BRV-C contribute to this activity. However, there was a strong difference in silencing activity between deletions that retained 441 and 183 bp of BRV-C, suggesting the presence of significant silencing activity within this 258 bp region. Sequences at both ends of this 258 bp region have almost complete conservation of nucleotide identity across 12 *Drosophila* species (*Figure 7C*), including two predicted binding sites for the transcriptional co-repressor Brinker (Brk). We generated a BRV-BC enhancer with both Brk sites mutated but found no loss of silencing activity (*Figure 7—figure supplement 1B–C*), indicating either that Brk does not function in mediating the repression, or that its activity is redundant with other mechanisms that are still functional. Thus, unrecognized motifs, likely within the conserved blocks clustered near the predicted Brk binding sites, must mediate the silencing activity associated with this region.

In many cases, developmentally-regulated gene silencing is mediated by the recruitment of Polycomb Group (PcG) proteins (*Schuettengruber et al., 2007*; *Schwartz and Pirrotta, 2007*), which act by promoting the trimethylation of histone H3 on lysine 27 (H3K27) (*Cao et al., 2002*). For many transgenes that contain elements capable of recruiting PcG proteins, silencing of the adjacent mini-*white* gene in the transgene is also observed (*Kassis and Brown, 2013*). In these cases, the silencing

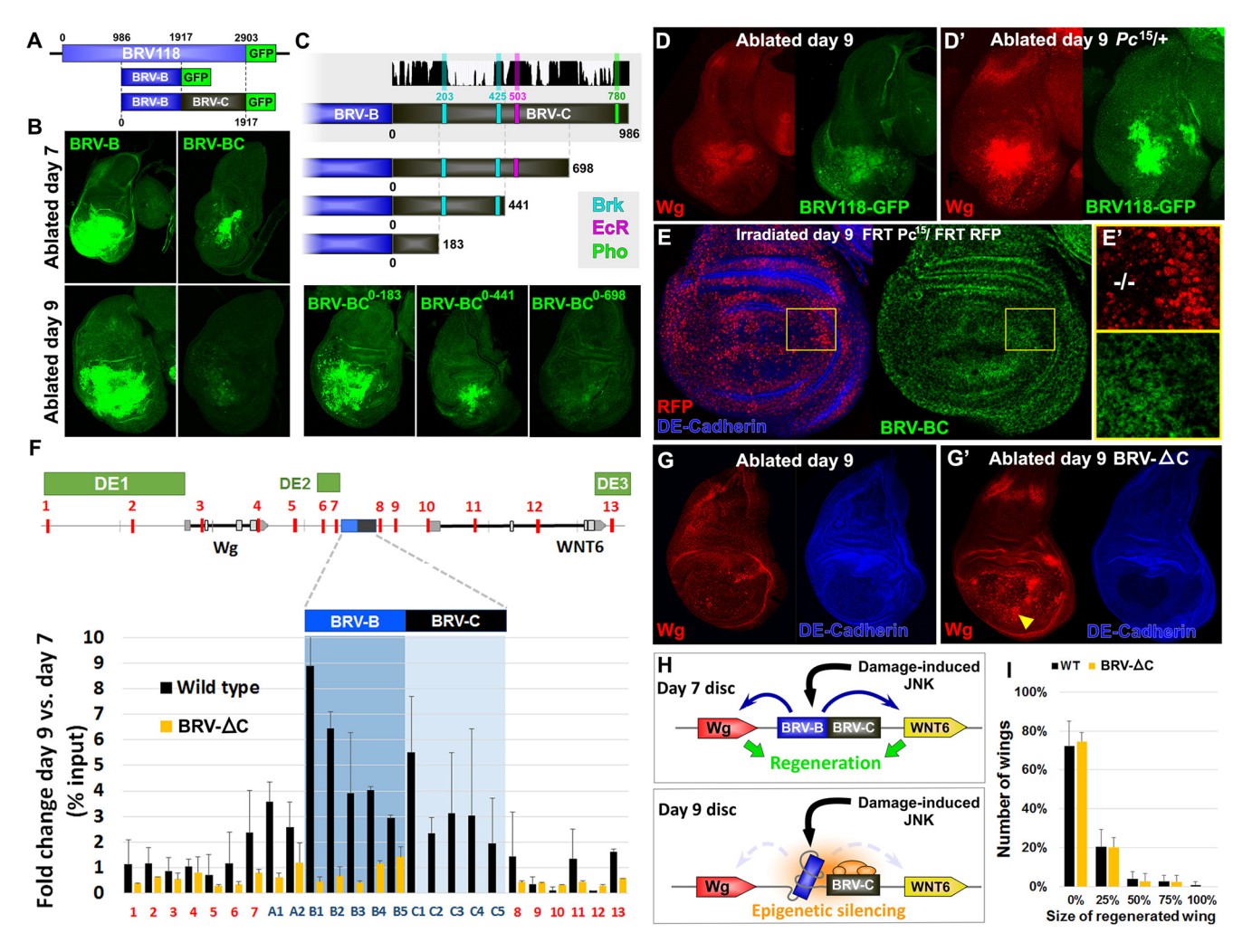

**Figure 7.** Polycomb-mediated epigenetic silencing of the BRV118 enhancer limits damage-responsive *wg* expression in mature discs. (**A**) Schematic of the BRV-BC reporter transgene in relation to BRV118 and BRV-B. Numbers indicate nucleotide positions. (**B**) Basal sections of day 7 discs bearing the BRV-B (left panels) or BRV-BC (right panels) reporters, ablated by *rn^ts^>egr* on day 7 (top row) or day 9 (bottom row). Addition of the BRV-C fragment reduces expression on day 7 and almost completely abolishes expression on day 9. (**C**) Schematic of the deletion series used to test silencing of BRV-B by BRV-C fragments (top). Predicted binding sites for Brinker (Brk), Ecdysone Receptor (EcR) and Pleiohomeotic (Pho) are shown as colored bars. The black histogram shows a measure of the evolutionary conservation in BRV-C, as in *Figure 6E*. Confocal sections of day 7 discs ablated by *rn^ts^>egr* bearing each BRV-BC derivative (bottom) shows that multiple regions across BRV-C are required for silencing of BRV-B. (**D-D'**) Day 9 discs bearing the BRV-BC reporter following *rn^ts^>egr* ablation in wild type (**D**) and *Pc^15^/+* heterozygotes (**D'**). Both damage-induced Wg (red) and reporter GFP expression (green) is elevated in *Pc^15^/+* discs compared to wild type. (**E-E'**) Heat shock induced *Pc^15^* mutant clones (marked by the absence of RFP) in a wandering stage larval disc bearing the BRV-BC reporter (green). Following clone induction, discs were irradiated with 45 Gy and dissected after 16 hr. BRV-BC reporter expression is elevated in *Pc^15^* clones (**E'**, -/- homozygous *Pc^15^*) but significantly lower in the wild-type twin spot (**E'**, red). (**F**) Chromatin immunoprecipitation by anti-H3K27me3 followed by qPCR to detect regions surrounding *wg* and *Wnt6* (numbered primer sets 1–13) and the BRV118 region (lettered primers sets). ChIP-qPCR was performed in wild type and homozygous BRV-ΔC day 7 and day 9 discs, and shown as the fold change between the two days. Schematic above shows numbered amplicon positions in the genome, WNT gene loci and previously identified developmental WNT enhancers (green, DE1: Wing disc hinge and embryonic enhancers, (*Neumann and Cohen, 1996*; *Von Ohlen and Hooper, 1997*), DE2: Notum and leg/antennal disc enhancer (*Pereira et al., 2006*), DE3: Eye/antennal disc and leg enhancer (*Koshikawa et al., 2015*)). In wild type discs (black bars) epigenetic silencing marks increase at the BRV118 locus in day 9 discs versus day 7, while levels at the *wg* and *Wnt6* coding sequences, and surrounding developmental *wg* enhancers, remain largely unchanged. Deletion of the BRV-C region of the enhancer from the genome abolishes the increase of H3K27me3 at the enhancer (yellow bars). Error bars are SD of repeats from 3 independent ChIP experiments. (**G-G'**) Day 9 wild type discs (**G**) or homozygous for BRV-ΔC (**G'**) following ablation by *rn^ts^>rpr*. Absence of the BRV-C enhancer fragment produces significantly more Wg in response to damage (**G'**, arrowhead). (**H**) Model of damage response regulation in a day 7 (top) and a day 9 (bottom) disc. JNK signaling activates *wg* and *Wnt6* expression in response to damage via the BRV-B region, but epigenetic silencing of the enhancer, nucleated by BRV-C, in a day 9 disc

*Figure 7 continued on next page*

*Figure 7 continued*

overrides this activation. (I) Assay of adult wing sizes that develop from homozygous BRV-ΔC discs ablated with *rn*^ts^>*rpr* on day 9. Absence of the BRV-C enhancer fragment does not alter regeneration of ablated disc. Error bars are SD of 3 biological repeats, n>200 flies per repeat.

The following figure supplements are available for figure 7:

**Figure supplement 1.** Additional characterization of the silencing activity in BRV118.

**Figure supplement 2.** Additional data on PcG-mediated repression by BRV-C.

**Figure supplement 3.** Cas9-mediated deletion of the BRV-C genomic region.

**Figure supplement 4.** A fragment of BRV-B driving *wg* rescues damage-induced *wg* expression in late stage discs, but does not improve regeneration.

of the mini-*white* gene is sensitive to the levels of PcG proteins. Indeed, we observed that transgenic flies bearing the BRV-BC reporter had lighter eyes than those bearing the BRV-B reporter (*Figure 7—figure supplement 2A*) even though both reporter genes were inserted at the identical site in the genome, suggesting that the BRV-C fragment was capable of mediating silencing of surrounding DNA when integrated at a genomic locus outside of the WNT cluster. However, in flies that were heterozygous for *pleiohomeotic* (*pho*) or *pleiohomeotic-like* (*phol*), the eye color was similar to those bearing the BRV-B fragment (*Figure 7—figure supplement 2A*), indicating that the silencing activity induced by the BRV-C fragment was indeed sensitive to PcG group gene dosage. We examined the sequence of BRV-C and found multiple potential binding sites for proteins shown to be important for PcG silencing at PREs, including 2 conserved *pho/phol* binding sites (*Figure 7—figure supplement 2B*). To test whether the silencing activity of BRV-C during disc regeneration is regulated by PcG, we examined the expression of the full length BRV118-GFP reporter in *rn*^ts^>*egr* imaginal discs heterozygous for the null allele of *Polycomb, Pc*[15]. In these flies, robust expression of the reporter was observed in day 9 discs, as was expression of the Wg protein expressed from the endogenous *wg* gene (*Figure 7D–D'*). However, due to changes in developmental timing, increased lethality, and the significant transdetermination levels associated with the *Pc* allele, documented previously (*Lee et al., 2005*), we were unable to test whether *Pc* gene dose affected levels of regeneration in our genetic ablation system. In order to avoid potential organism-wide pleiotropic effects of the *Pc* allele, we generated *Pc*[15] clones in developing discs that also had the BRV-BC reporter. Day 9 mosaic discs were irradiated to induce damage and examined for enhancer activity. Despite morphological disruption within the disc epithelium, likely due to the presence of *Pc* mutant tissue, and widespread cell death associated with irradiation, GFP was clearly expressed in the *Pc* mutant clones, and to a lesser extent in the surrounding heterozygous tissue, but was absent from the wild type sister clones (*Figure 7E–E'*). Thus, Pc is required to prevent damage-induced activation of the BRV-BC reporter in mature discs.

## Epigenetic silencing is restricted to the damage-responsive enhancer

To directly examine whether there is a change in the levels of PcG-mediated repression of the *wg* and *Wnt6* genes between day 7 and day 9, we used chromatin immunoprecipitation (ChIP) to examine the levels of H3K27 methylation across the WNT gene cluster from undamaged wing imaginal discs. In comparison to day 7 discs, there was a significant increase in the levels of H3K27 methylation in day 9 discs, which was mostly restricted to the BRV118 enhancer (*Figure 7F*). Most of the *wg* and *Wnt6* transcription units showed no change in H3K27 methylation, nor was there a significant increase in methylation observed in regions known to regulate aspects of developmental *wg* expression in third-instar discs (*Neumann and Cohen, 1996*; *Pereira et al., 2006*; *Koshikawa et al., 2015*) (*Figure 7F*). This is consistent with the observation that both *wg* and *Wnt6* are expressed in undamaged discs at this stage of development. Two other genomic regions examined, *act5C* and the heterochromatin *H23* locus, also showed no significant change between day 7 and day 9 (*Figure 7—figure supplement 2C*). Since the increase in the histone modification most characteristic of PcG-mediated silencing during L3 is mostly restricted to the BRV118 region, this suggests that the

damage-responsive enhancer is silenced in day 9 discs, while developmentally-regulated WNT enhancers are not.

To examine the importance of the silencing induced by BRV-C in vivo, we used CRISPR-mediated genome editing to generate a deletion that removed 1132bp, including the entire BRV-C fragment (BRV-ΔC, *Figure 7—figure supplement 3*). Flies homozygous for the deletion are viable and have no observable developmental abnormalities, but often hold their wings in an outstretched position reminiscent of the *wg^P* allele, in which the region encompassing the entire BRV118 enhancer is removed from the *wg* coding sequence by a genomic inversion (*Buratovich and Armer, 2002*). ChIP experiments performed on day 9 BRV-ΔC discs demonstrated a pattern of H3K27 methylation that resembles that normally found in wild-type day 7 discs in the region of the BRV118 enhancer (*Figure 7F*). Thus the sequences within BRV-C are necessary for the increased deposition of H3K27 histone marks by day 9. To examine the functional consequences of this changed pattern of H3K27 methylation, the level of *wg* expression was examined in regenerating *rn^ts>rpr* day 9 BRV-ΔC discs. In the absence of BRV-C, damage-induced *wg* expression is greatly increased in the disc proper epithelium surrounding the ablated pouch (*Figure 7G–G'*), indicating that the BRV-C region is required in vivo to suppress *wg* expression following damage in day 9 discs (*Figure 7H*). However, when assayed for regeneration, there was no observable difference between BRV-ΔC and flies bearing an intact enhancer region (*Figure 7I*).

We also attempted to restore expression of *wg* in another way. The BRV-B enhancer directs expression to a wide region surrounding the damaged tissue and only in response to tissue damage. Moreover, the BRV-B enhancer is equally active on day 7 and day 9. We therefore attempted to generate a transgene that expressed *wg* directly under the control of BRV-B, independently of Gal4/ UAS. However, we did not obtain viable transformants. We were able to obtain transformants where *wg* was driven by a sub-fragment of BRV-B, BRV-B3 (*Figure 7—figure supplement 4A*) possibly because it drives expression at lower levels (*Figure 7—figure supplement 4B*). While these transformants have high levels of *wg* following damage (*Figure 7—figure supplement 4C*), we did not observe an improvement in regeneration (*Figure 7—figure supplement 4D*) indicating that increased *wg* expression alone is insufficient to improve regeneration and that the damage-responsive expression of other genes necessary for regeneration might also be silenced in mature discs.

## *Myc* expression can reactivate the silenced enhancer and promote regeneration

*Myc* is upregulated during regeneration (*Figure 2B*, *Smith-Bolton et al., 2009*), and is therefore likely to be an important driver of regenerative growth. Moreover, we had previously shown that in immature discs, *Myc* promoted regeneration, while other growth promoting pathways tested, including CyclinD/CDK4, JAK/STAT and Rheb did not (*Smith-Bolton et al., 2009*). However, in those experiments, *Myc* was expressed under the control of Gal4/UAS, and it was therefore not possible to express *Myc* in cells that were not also expressing *egr*, and expression of *Myc* was confined to the time of ablation. To more rigorously test whether *Myc* could promote regeneration in older discs, we chose to target *Myc* expression to the blastema by directly driving expression under the control of BRV-B (*Figure 8A*). In contrast to our experience manipulating *wg* expression, we were able to obtain viable transformants.

In the absence of tissue damage, flies carrying the *BRV-B-Myc* transgene displayed no obvious growth abnormalities. However, following tissue ablation using *rn^ts>rpr* or *rn^ts>egr*, strong expression of *Myc* was detected even in day 9 discs (*Figure 8B*). In these discs, cells expressing the AP-1 reporter resembled the cells normally found in day 7 discs in that they were more flattened and *Mmp1*-expressing cells appeared to be migrating into the ablated portion of the disc (*Figure 8C–D*). Moreover, expression of Wg protein was increased in day 9 discs to levels comparable to those found in day 7 discs following ablation (*Figure 8E*). This likely results from an increase in *wg* transcription, since the full length BRV118 reporter, which includes the entire negatively acting BRV-C element, was also expressed in ablated day 9 discs (*Figure 8F*). This reactivation of the enhancer by *Myc* is cell autonomous, as clones expressing *Myc* in mature discs activate the enhancer following irradiation, while neighboring irradiated cells do not (*Figure 8G–G'*). Interestingly, clonal expression of *Myc* causes limited activation of the enhancer even in the absence of damage, but at a significantly reduced level than with irradiation (*Figure 8H–H'*). Thus, at least on day 9, expression of *Myc* is able to activate transcription mediated by an enhancer that is normally epigenetically silenced.

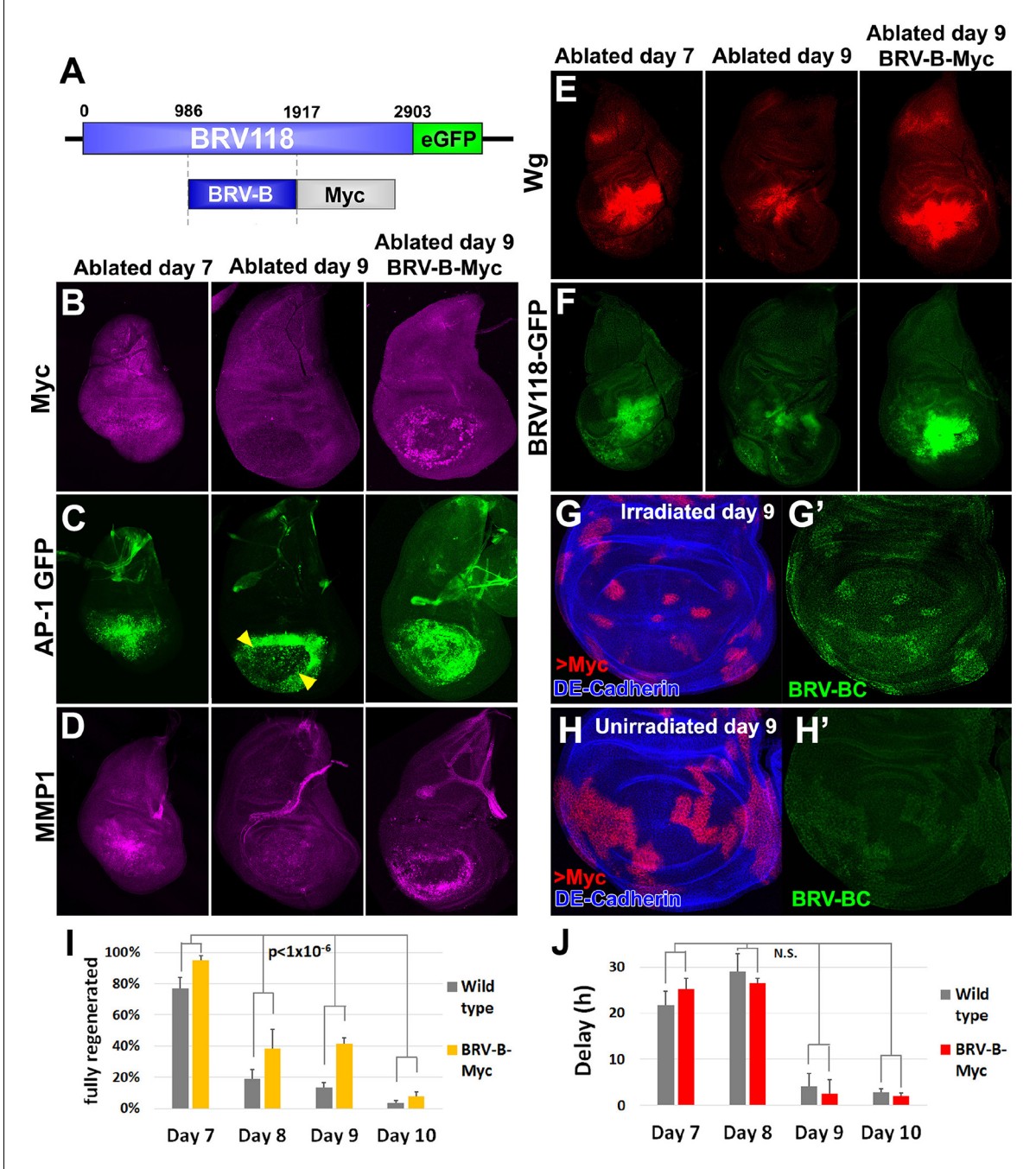

**Figure 8.** Circumventing the age-related repression of Myc improves regeneration. (A) Schematic of the *BRV-B-Myc* transgene. Numbers indicate nucleotide positions. (B-D) Basal sections of discs following *rn^ts^>rpr* ablation on day 7 (left panel) and day 9 (center panel), and on day 9 in discs bearing *BRV-B-Myc* transgene (right panel), imaged at the time of downshift. (B) Myc expression following ablation. Myc is expressed in basal wound periphery cells of a day 7 disc, but is absent in a day 9 ablated disc. The *BRV-B-Myc* transgene causes expression of Myc protein in an older disc. (C- D) Discs bearing the AP-1 GFP reporter (C) or stained with anti-Mmp1 (D), and imaged as in (B). AP-1 reporter expression is strongly expressed in basal cells of the pouch in day 7 ablated discs, coincident with Mmp1 expression. AP-1 reporter expression is found in the ring of wound periphery cells in day 9 ablated discs (arrowheads), while Mmp1 staining is reduced. Day 9 ablated discs bearing BRV-B-Myc have AP-1 GFP expressing cells covering the basal wound surface, resembling the expression pattern of day 7 discs, while Mmp1 expression is stronger than a wild type day 9 disc. (E-F) Discs stained for Wg (E) or bearing the BRV118-GFP reporter (F) following ablation with *rn^ts^>egr*, and imaged as in (B-D). Declining expression of the reporter on day 9 is rescued in discs bearing the *BRV-B-Myc* transgene. Similarly, *wg* expression is stronger on a day 9 bearing the *BRV-B-Myc* transgene, comparable to expression level in an ablated day 7 disc. (G-G') A day 9 disc bearing the BRV-BC enhancer with heat shock induced flip-out clones (marked by RFP, red) expressing *Myc*, irradiated with 45 Gy and dissected 16 hr later. Clones expressing *Myc* respond to damage by activating the

*Figure 8 continued on next page*

*Figure 8 continued*
BRV-BC enhancer (**G'**, green), while the enhancer remains inactive in neighboring damaged tissue not expressing *Myc*. (**H-H'**) Day 9 discs as in (**G-G'**), in the absence of irradiation. Expression of *Myc* in clones (red) causes low level of BRV-BC enhancer activity (**H'**, green), even in the absence of damage. (**I**) Assay of adult wing sizes that develop following ablation with *rn^ts^>rpr* on day 7, 8, 9 and 10 in wild type and *BRV-B-Myc* discs, quantifying animals that eclose with fully regenerated wings. The *BRV-B-Myc* expressing animals consistently eclose with more of the full regenerated wings compared to wild type. Mean differences between wild type and *BRV-B-Myc* data sets is statistically significant, (p-value calculated by two way ANOVA). Error bars are SEM of at least 3 biological repeats, scoring a total of >200 animals in each condition. (**J**) Developmental timing of wild type and *BRV-B-Myc* expressing larvae following *rn^ts^>rpr* ablation on day 7, 8, 9 and 10. Delay is measured as hours at which 50% of larvae have pupariated compared to unablated wild type. The *BRV-B-Myc* transgene (red bars) does not alter developmental timing following ablation compared to wild type (gray bars) on any day ablated. Mean differences between wild type and *BRV-B-Myc* data sets is not statistically significant (N.S.), calculated by two way ANOVA. Error bars are SEM of 3 biological repeats, n>100 flies for each genotype/condition.

Finally, as BRV-B driven *Myc* increases the expression of several genes that are usually absent from day 9 ablated discs, we examined whether regenerative ability was also affected. When compared to wild type, flies bearing *BRV-B-Myc* had significantly improved regeneration following ablation on day 7, 8, 9 and 10 (*Figure 8I*). This improvement in regeneration occurred without extending the delay in pupariation induced by ablation (*Figure 8J*). The improvement in regeneration elicited by *Myc* expression is modest by day 10, indicating that the loss of regenerative capacity becomes progressively more refractory to reversal even by increased levels of *Myc* expression. Overall, these data show that using an endogenous damage-responsive enhancer to target *Myc* expression specifically to cells surrounding the ablated tissue that give rise to the blastema can augment the regenerative process, likely by circumventing or reversing the repression of multiple genes that function in regeneration.

## Discussion

Many organisms lose the ability to regenerate damaged tissues as they mature (*Dent, 1962*; *Reginelli et al., 1995*; *Beck et al., 2003*; *Slack et al., 2004*; *Smith-Bolton et al., 2009*; *Porrello et al., 2011*; *Cox et al., 2014*). This change often occurs concurrently with a slowing of the growth of the organism, or a major transformation in its developmental state, e.g. metamorphosis in *Drosophila* (*Smith-Bolton et al., 2009*) and *Xenopus* (*Dent, 1962*). The loss of regenerative capacity is likely an important mechanism to balance the successful progression to reproductive adulthood at the cost of forming functionally complete tissue. Very few 'true' regeneration-specific genes have been identified (i.e. genes that are not required at any other time throughout the organism's life), but rather developmentally required pathways are often re-used during regeneration (*Sun and Irvine, 2014*). Thus, how regenerative growth can be selectively inhibited without compromising cell proliferation or differentiation remains unknown. Here we have shown that in the *Drosophila* wing disc this loss of regenerative capacity is achieved in part by the localized epigenetic inactivation of a damage-responsive enhancer that regulates the expression of *wg* and potentially *Wnt6*. This mechanism allows an organism to continue with its normal developmental program while shutting down its regenerative response to tissue damage.

Previous studies have demonstrated that the JNK pathway is robustly activated following tissue damage and has an important role in regenerative growth (*Bosch et al., 2005*; *Mattila et al., 2005*; *Bergantiños et al., 2010*). Our data confirm that JNK is strongly activated following damage, but furthermore, it appears similarly activated in both day 7 and day 9 discs, as assessed by the expression of an AP-1 reporter. Thus, the loss of regeneration that occurs between day 7 and day 9 cannot be attributed to a failure to activate JNK. Despite the similar levels of AP-1 activity, the cellular responses and changes in gene expression elicited by tissue damage differ considerably as the disc matures. Importantly, genes that are known to be downstream targets of the JNK/AP-1 pathway such as *Mmp1* (*Uhlirova and Bohmann, 2006*) and *DILP8* (*Colombani et al., 2012*; *Katsuyama et al., 2015*) have reduced expression on day 9 when compared to day 7. These changes in gene expression are likely to account for many of the differences in the cellular responses to tissue damage that we observe.

In addition to the aforementioned genes, the WNT genes *wg* and *Wnt6* also exhibit a significant decline in damage-induced expression with disc maturity. Our data shows this is due to the highly localized epigenetic silencing of a damage-responsive WNT enhancer, BRV118, that prevents their

expression specifically in response to injury in mature discs, but still allows expression from nearby developmentally regulated enhancers. This mechanism ensures that the contribution of both genes to a regeneration program can be shut off in mature tissues independently of their essential roles in growth and development of the disc. Our inability to detect expression of the BRV118-GFP reporter in unablated discs suggests that the BRV118 enhancer does not have a role in normal development. However, the *wg*[1] allele, which results in an incompletely-penetrant phenotype characterized by a failure to specify the wing pouch, is a deletion whose breakpoints are very close to the boundaries of the BRV118 fragment that we have studied (*Schubiger et al., 2010*). This suggests that a separate element, possibly very close to, but not fully contained within the boundaries of BRV118, may also be disrupted by the *wg*[1] deletion.

The expression profile of regenerating discs suggests the regulation of multiple genes is required during regeneration (*Klebes et al., 2005*; *Blanco et al., 2010*; *Katsuyama et al., 2015*), and that a significant number of these genes are also involved in developmental processes. Thus, equivalent regeneration specific enhancers, like BRV118, might also exist for these genes, such as *DILP8* and *Mmp1*. Both genes are known to be activated by JNK, although damage-responsive enhancers have not yet been characterized. Notably though, the *Mmp1*-lacZ reporter we used to investigate *Mmp1* activation (*Figure 2—figure supplement 2*), which accurately reflects Mmp1 protein expression following injury (*Uhlirova and Bohmann, 2006*), consists of a ~5 kb intronic region upstream of a lacZ reporter, which, based on its pattern of expression on days 7 and 9, must possess regulatory regions that allow both damage-induced activation and maturity-dependent silencing. Sequence comparison with BRV118 reveals several AP-1 binding sites that are identical to those found in BRV118, and multiple consensus sites for PcG repression. This combination of regulatory motifs could therefore reflect a molecular signature of genes that function in regeneration, and thus could potentially be used to identify genes that comprise a regeneration program through genome-wide analyses in the future.

Our studies of the regulation of *wg* expression have shown that, despite similar levels of JNK activation, increased levels of PcG-mediated epigenetic silencing can override the effect of JNK activation and suppress gene expression in late L3. PcG-mediated silencing is best characterized for its role in the epigenetic silencing of Hox genes during embryonic development in *Drosophila* (*Beuchle et al., 2001*; *Beck et al., 2010*), but also has important functions in imaginal disc development (*Pirrotta et al., 1995*; *Maurange and Paro, 2002*; *Perez et al., 2011*; *Mason-Suares et al., 2013*) and during regeneration (*Klebes et al., 2005*; *Lee et al., 2005*; *McClure et al., 2008*; *Katsuyama et al., 2015*). Indeed inappropriate cell fate switching following damage in imaginal discs (transdetermination) is associated with changes in PcG gene expression (*Klebes et al., 2005*; *Lee et al., 2005*; *McClure et al., 2008*), and in one instance JNK signaling reduced the extent of PcG mediated repression (*Lee et al., 2005*). A key property of epigenetic regulation by PcG is the ability to simultaneously silence multiple regions across the genome via the activity of a single master regulator complex, and, moreover, this silencing is heritable and thus its activation can maintain the locus in a repressed state through subsequent cell generations (*Déjardin and Cavalli, 2005*). Such a mechanism is ideally suited to the sustained and progressive silencing of a regeneration program during the ongoing growth and development of imaginal discs. However, unlike Hox genes, silencing of *wg* and *Wnt6* does not involve the entire transcription unit, but rather, is restricted to a damage-responsive enhancer. A similar local mode of epigenetic regulation has been described for the *Drosophila rpr* locus, in which epigenetic blocking of an irradiation-responsive enhancer region through enrichment of H3K27me3 prevents *rpr* expression following irradiation in late embryogenesis (*Zhang et al., 2008*). Importantly, the remainder of the *rpr* locus itself remains accessible, and is thus responsive to developmental signals required for programmed cell death to occur in the nervous system in late embryogenesis (*Maurange et al., 2008*; *Rogulja-Ortmann et al., 2008*). Localized epigenetic silencing of individual regulatory elements is therefore likely an important and potentially pervasive mechanism by which gene expression can be selectively activated or repressed by distinct inputs.

But how is this epigenetic silencing limited to just the enhancer? Elements that are responsible for expression of the "inner circle" of *wg* expression at the edge of the pouch and for expression in the leg disc are immediately adjacent to the BRV118 enhancer (*Pereira et al., 2006*). Thus, while the BRV-C fragment nucleates PcG-mediated repression that then spreads over the remainder of the BRV118 enhancer, mechanisms must exist that limit spread beyond the borders of the enhancer and

thus preserve the activity of the adjacent developmentally-regulated enhancers. Chromatin boundary elements that are able to block the spread of heterochromatin formation have previously been described (*Gaszner and Felsenfeld, 2006*) and are found in a variety of organisms including *Drosophila* (*Roseman et al., 1993*; *Kahn et al., 2006*; *Lin et al., 2011*). Unlike other boundary elements such as insulators that inhibit enhancer-promoter interactions (*Gurudatta and Corces, 2009*), these 'chromatin barrier' elements can prevent the propagation of repressive histone marks separately from a role in enhancer blocking (*Recillas-Targa et al., 2002*; *Lin et al., 2011*). Thus, a similar barrier element might be present within or near BRV118 to limit chromatin modifications to the damage responsive region, yet allow nearby developmental enhancers to remain active.

If multiple genes that function in regeneration have a similar bipartite mode of regulation, it is unlikely that expressing just one of these genes at a later stage of development can restore the ability to regenerate. Indeed, we found that restoring *wg* expression in day 9 discs did not promote regeneration. In contrast, expression of *Myc*, which is able to increase the levels of expression of both *wg* and *Mmp1*, and possibly the expression of other genes that are similarly regulated, was able to enhance regeneration. However, it is likely that Myc does not promote the expression of all genes that have been silenced in late L3. Indeed, unlike *wg* and *Mmp1*, we found that the JAK/STAT reporter is not reactivated in mature discs by the presence of Myc (data not shown). In addition, the delay in pupariation is not restored, which possibly results from a failure to restore the damage-responsive DILP8 expression level to that of a younger disc. While we have shown that Myc functions cell autonomously to reactivate BRV118-mediated expression of WNT genes, it is unclear whether Myc reverses the PcG-mediated repression of BRV118 or bypasses it completely. However, since the *BRV-B-Myc* transgene is only expressed in a small region of the disc, it is not easy to detect a change in the overall level of H3K27 methylation at the WNT locus in these cells with confidence. Additionally, even increasing Myc levels has little effect by day 10, suggesting that the silencing mechanism has become even more effective. It might be necessary to combine *Myc* overexpression with other manipulations to restore regeneration at even later stages. Previous studies have implicated Myc as a regulator of chromatin organization (*Amente et al., 2011*) and also as a regulator of cellular reprogramming (*Smith and Dalton, 2010*; *Chappell and Dalton, 2013*), and therefore studying the role of Myc in reactivating BRV118-mediated expression might provide a tractable way of understanding the role of Myc in these processes.

Overall, our investigation has revealed a mechanism by which genes required for both regeneration and development can be regulated to allow the age-dependent restriction of a regenerative response without affecting normal organismal growth and patterning of tissues. As PcG proteins are highly conserved from flies to vertebrates, as indeed are the targets they regulate (*Ringrose, 2007*), it would be of considerable interest to determine whether the loss of regenerative capacity in vertebrates also results from the selective epigenetic silencing of damage-responsive enhancers that regulate orthologs of *Drosophila* genes that we have studied, such as matrix metalloproteases and WNT genes.

## Materials and methods

### Fly stocks and ablation experiments

Stocks and crosses were maintained on yeast food at 25°C, except those for ablation experiments, which were maintained at 18°C. Stocks used in this study: $rn^{ts}>egr$ ($w^{1118}$;; *rn-Gal4, tub-Gal80$^{ts}$, UAS-egr*) and $rn^{ts}>rpr$ ($w^{1118}$;; *rn-Gal4, tub-Gal80$^{ts}$, UAS-rpr*) (*Smith-Bolton et al., 2009*), *AP-1-GFP* and *AP-1-RFP* reporters (*Chatterjee and Bohmann, 2012*), *Stat92E-GFP* (*Bach et al., 2007*), *bantam-GFP* (*Matakatsu and Blair, 2012*), *MMP1-LacZ* (*Uhlirova and Bohmann, 2006*), *PCNA-GFP* (*Thacker et al., 2003*), *CycE-GFP* (*Deb et al., 2008*), *puc-LacZ* (*Ring and Martinez Arias, 1993*), *nos-Cas9* (*Kondo and Ueda, 2013*), *vkg-GFP* (*Buszczak et al., 2007*), *dlg1$^{A40.2}$* (a gift from D Bilder). Stocks obtained from the Bloomington stock center: *wg$^1$* (BL2978), *UAS-hep$^{CA}$* (BL6406), *hep$^{r75}$* (BL6761), *UAS-yki* (BL28819), *Pc$^{15}$ FRT2A* (BL24468), *pho$^b$* (BL1140), *phol$^{81A}$* (BL24164), *DILP8-GFP* (BL33079), UAS-Myc (BL9674), Act5C>FRT.CD2>GAL4,UAS-RFP (BL30558), hs-Flp (BL8862). Genetic ablation experiments and scoring of adult wings was performed essentially as described in (*Smith-Bolton, 2009*) with each experimental condition compared to a suitable control that was

ablated and scored in parallel. Unless otherwise indicated, discs were dissected and fixed for immunofluorescence immediately after the ablation period.

## Generation of transgenic reporters and lines

The BRV118 enhancer was originally identified by S Carroll, and the BRV118-lacZ transgenic line that initiated this study was obtained from G Schubiger (*Schubiger et al., 2010*). The *BRV118-GFP* enhancer reporter was generated by cloning 2903 bp of the BRV118 genomic region upstream of the minimal *hsp70* promoter and e*GFP* coding sequence in p*EGFPattB* (K Basler). Reporter derivatives were generated by replacing the BRV118 enhancer DNA with the genomic regions listed in *Supplementary file 1A*. All GFP reporters were inserted into the *AttP40* landing site via PhiC31 recombination. The BRV-BΔAP1 and BRV-BCΔBrk transgenes were generated using InFusion PCR mutagenesis (Clontech, Mountain View, CA) to sequentially delete the consensus sequences (primers listed in *Supplementary file 1A*). The BRV-B-Myc transgene was generated by replacing the *eGFP* coding sequence in the BRV-B GFP reporter construct with the *Myc* coding sequence (GenBank: AY058627) and inserted into the *AttP40* landing site. The BRV-B3-Wg transgene was generated by replacing the *eGFP* coding sequence in BRV-B3 with the *wg* coding sequence (GenBank: BT133499) and 1079 bp of the 3'UTR to ensure proper transcript subcellular localization (*Simmonds et al., 2001*), and inserting into the *VK00013* (chr:3L) landing site. Cloning primers for *wg* and *Myc* constructs are listed in *Supplementary file 1*. Transgenic services were provided by BestGene (Chino Hills, CA).

## Immunohistochemistry

Discs were fixed and stained essentially as in (*Smith-Bolton et al., 2009*), and mounted in ProLong Gold Antifade Reagent (Cell Signaling, Beverly, MA). The following primary antibodies were used in this study: from the DSHB, Iowa City, IA; mouse anti-Wg (1:100, 4D4), mouse anti-Mmp1 (1:100, a combination of 14A3D2, 3A6B4 and 5H7B11), rat anti-DE-cadherin (1:100, DCAD2), mouse anti-Delta (1:100, C594.9B). Other antibodies; guinea pig anti-Myc (1:100, a gift from G. Morata), mouse anti-PHH3 (1:500, Cell signaling), rabbit anti-DCP-1 (1:250, Cell signaling), rabbit anti-GFP (1:500, Torrey Pines Laboratories, Secaucus, NJ), mouse anti-GFP (1:500 AB290, Abcam, Cambridge, MA), rabbit anti-β-galactosidase (1:1000, #559762; MP Biomedicals, Santa Ana, CA). Secondary antibodies used were from Cell Signaling, all at 1:500; donkey anti-mouse 555, donkey anti-rabbit 555, donkey anti-rat 647, donkey anti-rabbit 488 and donkey anti-mouse 488. Nuclear staining was by DAPI (1:1000, Cell Signaling). Samples were imaged on a Leica TCS SP2 Scanning confocal or Zeiss Light Sheet Z1.

To generate clonal patches of cells expressing *egr*, flies of genotype *hsflp; BRV-B; Gal80[ts], UAS-egr* were crossed to *act>>Gal4, UAS-RFP*. The progeny were heat shocked at 37°C for 15 min at 84 hr and 96 hr after egg deposition (AED) and maintained at 18°C to allow growth of Gal4 expressing clones. Larvae were transferred to 30°C at day 7 of development to inactivate Gal80[ts] and allow *egr* expression in clones. Discs and brain tissue were dissected after 24 hr *egr* expression, fixed and stained as described. To generate *Pc[15]* mutant clones, flies of genotype *hsflp; BRV-B; his::RFP FRT2A* were crossed to *Pc[15] FRT2A/TM6Csb*. The progeny were maintained at 18°C, heat shocked at 37C for 2 hr at 48 hr and 72 hr AED, and irradiated with 45 Gy on day 9. After 16 hr recovery, discs were dissected, fixed and stained as described. Control larvae were processed identically, omitting the irradiation step. To generate Myc expressing clones, flies of genotype *hsflp; BRV-BC; Act5C>FRT.CD2>Gal4,UAS-RFP* were crossed to UAS-Myc and heat shocked 48 hr AED for 10 min. Larvae were irradiated with 45 Gy on day 9 and dissected as above.

## Chromatin immunoprecipitation and qPCR

For each immunoprecipitation (IP) chromatin was prepared from approximately 300 wing discs from day 7 larvae or 200 wing discs from day 9 larvae, and the IP was performed essentially as follows: larvae were dissected in small batches in ice cold Schneider's medium, fixed in 1% paraformaldehyde in PBS for 10 min at room temperature, the reaction was quenched with 250 mM glycine, washed in PBS three times for 10 min at room temperature, discs were removed from carcasses in lysis buffer (10 mM Tris HCl pH8, 1 mM EDTA, 0.1% SDS, 1 mM PMSF, 1% Triton-X100, Complete protease inhibitor [Roche Diagnostics, Indianapolis, IN]), pelleted and snap frozen in liquid nitrogen. Pellets

were pooled and resuspended in 500 ul lysis buffer, sonicated with a Biorupter (Diagenode, Denville, NJ) on high for 12.5 min, 30 s on 30 s off in ice to yield chromatin in the 500–1000 bp range. The sonicated samples were placed into RIPA buffer (lysis buffer plus 10% DOC and 140 mM NaCl), centrifuged on max for 5 min at 4o°C, and transferred to a new tube. 60 µl was removed for ethanol precipitation and run on a 1.5% gel to test sonication efficiency. From the remaining chromatin sample 5% was removed as input, and the rest was used for IP with 10 µg anti-H3K27Me3 (EMD Millipore, Billerica, MA) in RIPA buffer with overnight incubation at 4°C, or no antibody control. IP samples were purified on Protein-A sepharose 4B beads (Sigma-Aldrich, St. Louis, MO) in RIPA buffer for 30 min at room temperature. Beads were washed four times in RIPA buffer, once in RIPA buffer with 500 mM NaCl, once in LiCl2 ChIP buffer (10 mM Tris HCl pH8, 1 mM EDTA pH8, 1% DOC, 1% NP40, 250 mM LiCl2) and three times in TE buffer (10 mM Tris HCl pH8, 1 mM EDTA pH8) for 15 min each wash. Beads were resuspended in 50 ug/ml RNAseA (New England Biolabs, Ipswich, MA) for 30 min incubation at 37°C, washed in TE, and resuspended in elution buffer (0.1M NaHCO3 pH10, 1% SDS). Beads were eluted twice, 15 min each and the eluate combined in a new tube with 40 mM Tris HCl pH8, 250 mM NaCl, 10 mM EDTA and 0.5 mg/ml Proteinase K (New England Biolabs). Samples were incubated at 45°°C for 3 hr for proteinase digestion, then at 65°C overnight to reverse crosslinking, followed by ethanol precipitation. Input samples were also RNAse A and Proteinase K treated and purified as for IP samples. Chromatin was purified, diluted 1:10 to 1:100 for qPCR, which was performed on a StepOnePlus qPCR System (Life Technologies, Carlsbad, CA) using primers listed in *Supplementary file 1B*. Each IP was performed 3 times from independent dissections, while qPCR amplifications were repeated a minimum of 3 times on each chromatin sample.

## CRISPR/Cas9 generation of mutants

The FlyCas9 system (*Kondo and Ueda, 2013*) was used to generate the BRV-ΔC genomic deletion. The following annealed primer pairs were cloned into pU6.2B vector: 5'-CTTCGAATCGCCCGCTC-AGACAGT-3' and 5'-AAACACTGTCTGAGCGGGCGATTC-3', and 5'-CTTCGGCTTTCTGCTATTG TTGCT-3' and 5'-AAACAGCAACAATAGCAGAAAGCC-3'. The plasmid was inserted into the AttP2 landing site by PhiC31 recombination, yielding a stable transgenic line expressing 2 guide RNAs that target the 5' and 3' ends of BRV-C. This line was crossed to a line expressing germline Cas9, as described in the mating scheme (Figure 7—figure supplement 3A), and potential deletions identified by genomic PCR screening.

## Ex vivo culture

Discs from day 7 or day 9 larvae bearing the BRV-B reporter were dissected and wounded by cutting a fragment from the posterior ventral region with tungsten wires. After wounding, discs were cultured in either Schneider's or Robb's medium (*Robb, 1969*) supplemented with 10% fetal bovine serum and 1% penicillin/streptomycin for 12 hr on a Nunc Lab-Tek II chamber slide, (Thermo Fisher Scientific, Waltham, MA), before being fixed and stained. To examine BRV-B activation in the absence of JNK activity the small molecule JNK inhibitor SP600125 (*Bennett et al., 2001*) was dissolved in 1% DMSO to prevent precipitation in the culture medium, and added at a final concentration of 10 µM. Control discs were cultured in 1% DMSO in culture medium. Discs were visualized directly on a Zeiss Axio Imager M1 without fixation or antibody staining.

## Irradiation

Density controlled wandering stage larvae were placed on shallow yeast food and irradiated with 45 Gy in an X-ray cabinet (Faxitron, Tucson, AZ), followed by recovery for 16 hr before dissection.

## RNA in situ hybridization

Discs were dissected and fixed as for immunofluorescence, and RNA *in situs* were performed according to established methods for alkaline phosphatase based dig-labelled probe detection. Digoxigenin labelled antisense probes were generated targeting the WNT gene coding sequences using the following primer pairs: WNT4: forward 5'-AGTCGAGTGCCGAACGAGCTG-3' and reverse 5'-TTGT-AAAGGCCTTGCTGCATATCCATGT-3', WNT6: forward 5'-ATTCCCGAGAGACGGGTTTCGTG-3' and reverse 5'-GCGACTATTTACAAGGCTTACATGAGC-3', WNT10: forward 5'-GCTACCGAGAGA-GTGCGTTCGC-3' and reverse 5'-CCTGTATATCAGCTCCCAGATCGCG-3', Wg: forward 5'-

CAACAGTCTCCGGGCCACCAAC-3' and reverse 5'-CGATTGCATTCGCATTTTTCTGCTCGC-3'. Sense probes were generated using the same DNA sequences, and *in situs* were performed to ensure specificity. Control and experimental discs were stained simultaneously for the same duration, mounted in Permount (Fisher Scientific, Pittsburg, PA) and imaged on a Zeiss Axio Imager M1.

## Gut experiments

Newly eclosed adults were cultured in a vial with filter paper soaked with 5% sucrose and 5% dextran or 5% dextran sodium sulphate (DSS, MP Biomedicals) in water as a food source. Adults were incubated at 30°C for two days, and filter paper was replaced every 12 hr. Males were dissected and gut tissue was fixed and stained, as in (*Amcheslavsky et al., 2009*).

## Developmental timing assay

Larvae were density controlled, with 50 larvae to a yeast food vial supplemented with yeast paste, and aged at 18C. Larvae were scored for puparium formation every 12 hr from the beginning of the temperature upshift that activates ablation, and throughout the regeneration period in which the larvae were maintained at 18°C. At least 3 independent ablation experiments (separate upshifts) using multiple vials (>250 larvae per repeat) were used to generate pupariation curves, which were plotted to estimate the average delay of puparation following damage.

## Acknowledgements

The authors would like to thank D Bilder, A Dernburg, M Welch, D Bohmann, G Morata, A Teleman and G Schubiger for stocks, reagents and equipment. We thank the Bilder and Hariharan lab members for critical reading of the manuscript and advice. We thank S Reich, H Aaron and A Strom for technical assistance, and J Fristrom for disc culture expertise. We thank the Bloomington Stock Center, Drosophila Genomics Resource Center, Developmental Studies Hybridoma Bank, BestGene and AddGene, and the UC-Berkeley Molecular Imaging Center for stocks, reagents and services.

## Additional information

### Funding

| Funder | Grant reference number | Author |
|---|---|---|
| National Institutes of Health | R01 GM085576 | Iswar K Hariharan |
| American Cancer Society | Research Professor Award, 120366-RP-11-078-01-DDC | Iswar K Hariharan |
| California Institute of Regenerative Medicine | Postdoctoral Fellowship | Robin E Harris |
| Life Sciences Research Foundation | Postdoctoral Fellowship | Robin E Harris |

The funders had no role in study design, data collection and interpretation, or the decision to submit the work for publication.

### Author contributions

REH, LS, Conception and design, Acquisition of data, Analysis and interpretation of data, Drafting or revising the article; JS, Acquisition of data, Analysis and interpretation of data; IKH, Conception and design, Analysis and interpretation of data, Drafting or revising the article

### Author ORCIDs

Iswar K Hariharan, http://orcid.org/0000-0001-6505-0744

## Additional files

### Supplementary files

• Supplementary file 1. Cloning and qPCR primers tables. (A) Primers used for cloning transgenic reporters of the BRV118 enhancer and all derived constructs. (B) Primers used for qPCR analysis of ChIP DNA (*Figure 7F* and *Figure 7—figure supplement 2C*).

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
