## [Decision Letter]

Thank you for submitting your work entitled "Epigenetic silencing of a damage-responsive WNT enhancer limits regeneration in maturing *Drosophila* imaginal discs" for consideration by *eLife*. Your article has been reviewed by three peer reviewers, and the evaluation has been overseen by Hugo Bellen as the Reviewing Editor and K VijayRaghavan as the Senior Editor. One of the three reviewers, Richard Mann, has agreed to share his identity.

The reviewers have discussed the reviews with one another and the Reviewing editor has drafted this decision to help you prepare a revised submission..

Summary:

This manuscript describes an impressive and exciting study from the Hariharan lab of the function of an apparently regeneration-specific enhancer in promoting regeneration of imaginal discs in *Drosophila*. The work here significantly advances previous studies of disc regeneration carried out by the Hariharan lab and others and demonstrates that silencing of a specific enhancer plays a major role in limiting a regeneration response to tissue damage. Since many organisms lose the ability to regenerate with increasing age this work suggests a mechanism that may be broadly used.

Using a conditional system of inducing tissue damage and regeneration the authors explore their previous finding of differences in the regeneration response between damage induced "early"(day 7) versus "late" (day 9), which correlated with ability of Wg to be activated. Here they find that Wg expression during regeneration requires activation of the BRV-118 enhancer, first identified by the late Gerold Schubiger. This enhancer is located at the Wg/Wnt locus and regulates both Wg and *Wnt6*, and its activation occurs after early damage but not late damage. Structure function studies indicate that the enhancer has separate activating and repressive components and consists of an activating region with AP1 sites (among others) and a repressive element that contains PcG binding sites; these factors are demonstrated to be genetically required for the activation and repression, respectively. They observe increased H3K27me, a marker of PcG silencing, on the enhancer after damage at day 9, but not day 7; but none on other regions of *Wnt6* or Wg genes nor on developmental Wg enhancers in L3 during normal development, suggesting selective silencing of this damage-responsive enhancer.

The authors then use genome editing to delete the repressive region of the enhancer. The resulting flies are viable, and develop relatively normally (except for a held-out wing phenotype like wgP, which also correlates with removal of the enhancer region). H3K27me is absent from BRV-118 region in these flies, consistent with a requirement for the repressive sequences in deposition of H3K27me. When damaged at day 9, however, regeneration was not enhanced over controls with an intact BRV-118 even though Wg expression was induced in the discs similar to an early damage response. However when Myc was driven by BRV-B, Wg, MMP1 and BRV-118:GFP enhancer induction was increased after damage at day 9 and regeneration was improved ~20% over controls. Since the delay was not increased it appears that Myc expression increases the rate of recover, although not sufficient to account for all aspects of regeneration.

Essential revisions:

The authors argue that theBRV-118 enhancer is specific to the damage response, stating that it is not expressed in L3 discs under normal conditions. However, I wonder if the authors should consider the idea that the enhancer could be used as a "developmental" enhancer earlier than they have looked. A few of their statements suggest it is worth looking into: they suggest that the *wg^[37]^* phenotype is due to less robust wing pouch specification; they observe that BRV-118 is not relevant to Wg expression in the notum even after damage; their "early L3" disc is actually not so early, as the Wg pattern is mature in the discs shown; and they make the point that BRV-118:GFP is not expressed in the absence of damage in early L3 wing discs. However, we do not get to see if it is expressed at any earlier stage. What does it look like prior to the maturation of the Wg pattern of expression (at 25^o^C this would correspond to prior to 80-85 hrs AEL)? Is BRV-118 "only" for regeneration, or is it an early (perhaps transient) enhancer that boosts Wg to promote robust wing pouch specification, that has been co-opted for use in regeneration (which also needs a boost)? If so, this would be consistent with and provide support for the view that disc regeneration involves a reversion to earlier developmental state.

*Myc* can induce cellular growth and proliferation. Can other inducers of these changes in cellular behavior also rescue regeneration in older discs? In other words, how specific is *Myc*'s ability to do this? Is the *myc* rescue of regeneration cell autonomous?

---

## [Author Response]

*Essential revisions:*

*The authors argue that theBRV-118 enhancer is specific to the damage response, stating that it is not expressed in L3 discs under normal conditions. However, I wonder if the authors should consider the idea that the enhancer could be used as a "developmental" enhancer earlier than they have looked. A few of their statements suggest it is worth looking into: they suggest that the wg^[37]^ phenotype is due to less robust wing pouch specification; they observe that BRV-118 is not relevant to Wg expression in the notum even after damage; their "early L3" disc is actually not so early, as the Wg pattern is mature in the discs shown; and they make the point that BRV-118:GFP is not expressed in the absence of damage in early L3 wing discs. However, we do not get to see if it is expressed at any earlier stage. What does it look like prior to the maturation of the Wg pattern of expression (at 25^o^C this would correspond to prior to 80-85 hrs AEL)? Is BRV-118 "only" for regeneration, or is it an early (perhaps transient) enhancer that boosts Wg to promote robust wing pouch specification, that has been co-opted for use in regeneration (which also needs a boost)? If so, this would be consistent with and provide support for the view that disc regeneration involves a reversion to earlier developmental state.*

Thanks for raising this point. We have put in a considerable amount of effort into addressing this issue. Thus far, we can find no evidence that this enhancer is expressed above background levels at any stage of disc development. In the revised manuscript, we have included an image from an L2 disc where wingless expression is still in the immature diffuse pattern in the pouch and which shows a clear lack of reporter expression. We have also examined discs from the L1 stage (not easy to dissect and mount) but they do not have any reporter expression above background either.

This then raises the issue of why the *wg^[37]^* allele, which has a deletion that approximates the boundaries of the BRV118 fragment we have studied, has an incompletely penetrant phenotype characterized by a defect in wing pouch specification. One possibility is that the *wg^[37]^* chromosome has a second mutation that is responsible for this phenotype. We have excluded this possibility – we have generated the same deletion using CRISPR-Cas9 on a “clean chromosome” and found that those flies also have the same incompletely-penetrant wing-specification phenotype. We suspect that there might be a developmentally-regulated enhancer that functions early in larval development that is close to the boundary of the *wg^[37]^* deletion and which may be disrupted by this deletion. However, this enhancer is likely to be distinct from the damage-responsive enhancer since the BRV-118 reporter which contains a robust damage-responsive enhancer is not expressed in the absence of damage. We have included a sentence to summarize this conclusion in the revised manuscript.

*Myc can induce cellular growth and proliferation. Can other inducers of these changes in cellular behavior also rescue regeneration in older discs? In other words, how specific is Myc's ability to do this? Is the myc rescue of regeneration cell autonomous?*

These are all good points. In the revised manuscript, we have included some additional experimental data that show that reactivation of the enhancer in older imaginal discs appears to be cell autonomous.

In previous work from our laboratory (Smith-Bolton et al. 2009) we had shown that even in younger discs, *Myc* was more effective at promoting regenerative growth than other growth drivers that we tested (Rheb, CyclinD/CDK4, JAK/STAT). In subsequent unpublished experiments using older discs, we obtained results that hinted that *Myc* might also promote some regeneration in older discs. However in all these experiments, we expressed the growth-promoting gene under UAS control. It was therefore expressed in the ablated cells and in some cells that expressed the proapoptotic gene yet survived the ablation. In this study, we used the BRV-B fragment to target expression to the blastema. Since the earlier experiments suggested that other growth-promoting genes were ineffective at promoting regeneration even in younger discs, we chose to focus our efforts on whether *Myc* could promote regeneration in older discs. Hence when we found a way to target expression to the regeneration blastema using the BRV-B fragment, we only generated BRV-B-driven Myc transgenics and did not pursue the other pathways.

In response to the reviewers’ suggestion, we came up with a way of testing whether re-activation of the enhancer by *Myc* in older discs is cell autonomous. This is difficult to do using the genetic ablation system. However, the BRV-BC enhancer is also robustly activated by X-irradiation in younger (day 7) discs but is silenced in day 9 discs. By generating FLP-out clones that express UAS-myc, we now show clearly that the re-activation of the enhancer is cell-autonomous. Interestingly, we find that *Myc* activates the silenced enhancer, albeit at a much lower level, in day 9 discs even in the absence of radiation. These data have been included in Figure 8 in the revised manuscript.